# LEARNING TO INTERRUPT IN LANGUAGE-BASED MULTI-AGENT COMMUNICATION

## ABSTRACT

Multi-agent systems using large language models (LLMs) have demonstrated impressive capabilities across various domains. However, current agent communication suffers from verbose output that overload context and increase computational costs. Although existing approaches focus on compressing the message from the speaker side, they struggle to adapt to different listeners and identify relevant information. An effective way in human communication is to allow the listener to interrupt and express their opinion or ask for clarification. Motivated by this, we propose an interruptible communication framework that allows the agent who is listening to interrupt the current speaker. Through prompting experiments, we find that current LLMs are often overconfident and interrupt before receiving enough information. Therefore, we propose a learning method which predicts the appropriate interruption points based on the estimated future reward and cost. We evaluate our framework across various multi-agent scenarios, including 2-agent text pictionary games, 3-agent meeting scheduling and 3-agent debate. Experiment results show that our HANDRAISER can reduce communication cost by 32.2% compared with the baseline with a comparable or superior task performance. Such learned interruption behavior can also generalize to different agents and tasks.

## 1 INTRODUCTION

Large language model (LLM)-based multi-agent systems have shown remarkable performance in various domains, including *reasoning* (Du et al., 2023; Zhuge et al., 2024; Qian et al., 2025), *software engineering* (Hu et al., 2025; He et al., 2025). They also show potential in simulated social environments such as *strategic games* (Xu et al., 2023; Wang et al., 2023) and *social behavior modeling* (Park et al., 2023; Dubois et al., 2023). Compared with single-agent systems that rely on a single LLM to complete a task, multi-agent systems employ LLMs built with potentially different capabilities, information access, efficiency, and even objectives to communicate and collaborate. Despite their effectiveness and potential, multi-agent systems typically suffer from fast-growing context due to the verbosity of typical LLM-generated messages, as well as the number of messages being sent or broadcast across different agents. Such inefficient communication not only decreases the general performance for each agent due to overloaded context (Guo et al., 2024; Cemri et al., 2025), but also increases the amount of compute and incurs larger latency at inference time (Wang et al., 2025; Zhang et al., 2025).

Consider a single interaction (*i.e.,* message) in a multi-agent system. It involves two types of roles: the speaker who sends the message, and the listener who receives it; and efficient communication should aim to convey the speaker's intent to the listener with minimal words. Towards efficient multi-agent communication, previous work mostly focuses on prompting or training the speaker to generate more succinct messages. This can be seen as speaker-oriented compression (Fang et al., 2025; Qiao et al., 2025). While such speaker-oriented compression is initially helpful in reducing overall verbosity, further compression becomes significantly challenging. It requires the speaker to identify the parts that are indispensable to the listener. More importantly, such parts may also vary for different listeners. Take the "Text Pictionary" game as an example. In this game, one agent describes a secret entity and the other agent guesses the answer. As shown in Fig. 1a, given the same 16-word description, some agents can already guess the word while other agents are still struggling.

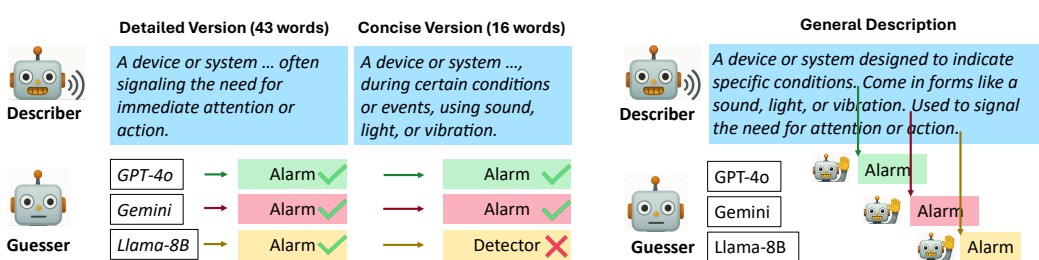

(a) The speaker(describer)-oriented compression can not generalize across different listeners (guessers).

(b) We let the listeners (guessers) themselves to decide when to interrupt and respond.

Figure 1: Text Pictionary game as an example: a describer describes a word for the guesser to guess.

In this work, we introduce an *interruptible communication framework* for the multi-agent LLM systems. For each interaction, the current "speaking" agent sends a stream of tokens to the listener. The listener will determine whether to interrupt the speaker. This allows it to skip information it deemed redundant or ask for clarification when confused. Once decided to interrupt, the listener first sends an interruption signal. Then it starts to generate a response. Upon receiving the signal, the speaker halts the generation process and awaits the listener's response. Such an interruption also naturally occurs in human conversation. People can interrupt each other to accelerate information exchange and make the communication more efficient (Lycan, 1977; Bennett, 1978; Ng et al., 1995). In contrast to speaker-oriented communication in Fig. 1a, the interruption mechanism shown in Fig. 1b allows different listeners (*i.e.,*, guessers) to interrupt at different points. They can interrupt when they are ready to make a guess. There is no need for the speaker (ı,e describer) to tailor its messages.

To teach the LLMs when to interrupt, we investigate both prompting-based methods and training-based methods. During prompting experiments, we find that the LLM agents are often overly confident on their understanding level and eager to interrupt prematurely. Therefore, we sampled multi-agent rollouts and post-annotated each potential interruption point based on its *payoff* compared to not interrupting. To estimate the payoff, we use tree sampling to estimate the expectation of potential token reductions, as well as any drop of final task performance. With such training data, the agent is finetuned to learn to predict the points with a higher task reward and a lower communication cost to interrupt. We evaluate our interruptible communication framework and trained HANDRAISER model with multiple LLMs (`Llama-3.1`, `GPT-4o`, and `Gemini-2.0-flash`) across three multi-agent scenarios from both simulated games and real-world tasks. Experimental results show that our HANDRAISER can reduce the communication cost by 24.3% on textual pictionary, 23.4% on meeting scheduling, and 48.9% on multi-agent debate compared with the generic baseline. Further analysis show that the interruption behaviors learned by HANDRAISER can be transferred to other types of speaking agents and tasks.

## 2 INTERRUPTIBLE LANGUAGE-BASED COMMUNICATION

In this section, we introduce an interruptible communication framework that operates through a listener-oriented decision process, which dynamically evaluates whether to stop the ongoing generation of the speaker and start responding.

### 2.1 INTERRUPTIBLE COMMUNICATION PROTOCOL

**Basic formulation.** Consider one back-and-forth round of interaction illustrated in Fig. 2, with Alice $\mathcal{A}$ speaking first and Bob $\mathcal{B}$ responding afterwards. With current multi-agent communication framework (illustrated on the left), Bob is simply waiting on Alice to finish while Alice is "speaking", and only after Alice finished generating all the tokens and send the full message to Bob can Bob start the encoding and generation process. In our proposed interruptible communication protocol (illustrated on the right of Fig. 2), the speaker Alice generates its response $X_t$ in fixed-sized

Figure 2: In non-interruptible communication (left), one agent (Bob) must wait for completion of full generation from another agent (Alice) to start responding. However, in the interruptible case, the speaker Alice sends messages to the listener Bob in chunks, and Bob will decide whether to interrupt. Upon interruption, Alice halts the generation and Bob starts to respond.

chunks[1], *i.e.,* $X^t = \{c_0^t, \cdots, c_n^t\}$, and each chunk $c_i^t$ is transmitted to the listener after they are generated. Upon receiving each chunk $c_i^t$, the listener Bob decides whether it wants to interrupt based on the previous chat history, as well as the prefix of the message by accumulating all the chunks it received so far in this round, *i.e.,* $\hat{X}^t = \{c_0^t, \cdots c_i^t\}$. And when Bob decides to interrupt, it sends an interruption signal to Alice and begins generating its own response. Simultaneously for Alice, upon receiving this signal, halts its current generation process and awaits Bob's response.

**Generalizing to more than two agents.** Such an interruptible communication protocol also generalizes beyond 2-agent scenarios. Consider a $k$-way conversation engaged in a predetermined order (*e.g.,* round-robin), instead of waiting the current speaking agent $\mathcal{A}_i$ to finish generating, the next agent in line $\mathcal{A}_{i+1}$ receives chunks of the message while $\mathcal{A}_i$ is generating and decide whether to interrupt and generate its response earlier. When communicating without any predetermined order (*e.g.,* as a "group chat"), the message from the speaking agent $\mathcal{A}_i$ is *broadcasted* to all other agents as they are independently deciding whether to interrupt. [2]

This interruptible mechanism can improve the communication efficiency in two ways: (i) The generation cost of Alice can be saved once its generation is halted; and (ii) the communication latency is reduced because Bob does not need to wait the full message and can potentially respond earlier. We discuss more on the payoffs of an interruption in the next section.

## 2.2 EVALUATING THE PAYOFF OF INTERRUPTIONS

Akin to the compression-fidelity tradeoff in information theory (Shannon et al., 1959), an interruption in communication also constitutes a tradeoff between communication cost and quality, for which we measure as total number of generated tokens and final task performance, respectively.

**Estimating communication cost and quality.** Given a set of $m$ agents where $A = \{a_1, \cdots, a_m\}$, a conversation $C_A^{1:T} = \{X_{a_1}^1, X_{a_2}^1, \cdots, X_{a_m}^1, \cdots, X_{a_m}^T\}$ denotes $T$ rounds of messages where each message $X_a^t$ is the message generated by agent $a$ at round $t$. Following previous work in optimizing multi-agent communication , we estimate the communication cost by counting the total amount of tokens generated by all agents till the end of the conversation, *i.e.,* $Cost(C_A^{1:T}) = \sum_{a \in A} \sum_{t=1}^{T} |X_a^t|$. And to measure the overall communication quality, we use goal-oriented task performance of respective tasks, *i.e.,* $Perf(C_A^{1:T}) \in [0, 1]$, to measure the communication quality.

**Balancing the cost-quality tradeoff.** When considering whether it is ideal to interrupt after receiving a certain chunk $c_i^t$ for agent $a$, it is important to note that it not only affects the current turn $X_a^t$ at round $t$, but also every turn afterwards. This is because the truncated message $\hat{X}_a^t$ from agent $a$ will be factorized as the context for all subsequent interactions $\hat{C}_A^{t:\hat{T}}$. It may even end at a different round $\hat{T}$ depending on the ending criteria. Thus for the interruption of agent $a$ at $t$-th round

---

[1] We use the notion of chunks to strike a balance between the granularity of interruption points and amount of interruption predictions needed. And we treat their sizes (a single token $\sim$ full message) as a hyperparameter.

[2] While contention might exist as two agents might both want to interrupt, we can refer to common resolutions of this, such as first-come-first-served.

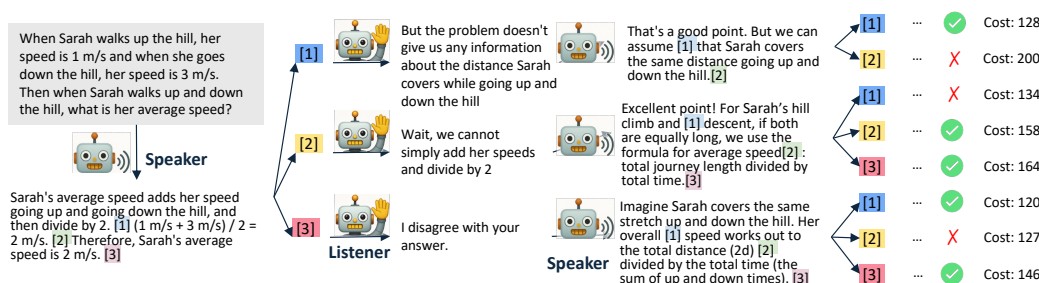

Figure 3: Tree-sampling to estimate the expectation of comm. cost and final task performance.

after its $i$-th chunk, we formulate the potential changes (*i.e.*, $\Delta$) in cost of communication and task performance as:[3]

$$\Delta_{Cost}(a, t, i) = -\sum_{k=i+1}^{n} |c_k^t| + \mathbb{E}[Cost(\hat{C}_A^{t:\hat{T}})] - \mathbb{E}[Cost(C_A^{t:T})] \tag{1}$$

$$\Delta_{Perf}(a, t, i) = \mathbb{E}[Perf([C_A^{1:t-1} \parallel \hat{X}_a^t \parallel \hat{C}_A^{t:\hat{T}}])] - \mathbb{E}[Perf(C_A^{1:T})] \tag{2}$$

where $\parallel$ denotes the concatenation of turns. An ideal interruption would aim to reduce communication cost, *i.e.*, $\min \Delta_{Cost}(a, t, i)$, while maximizing the task performance, *i.e.*, $\max \Delta_{Perf}(a, t, i)$. And to combine these two objectives, we consider an interruption to have *positive payoff* when it reduces the communication cost while maintaining the task performance, *i.e.*, $\Delta_{Cost}(a, t, i) < 0 \land \Delta_{Perf}(a, t, i) \geq 0$.

## 3 LEARNING TO INTERRUPT WITH LANGUAGE MODELS

### 3.1 PROMPTING-BASED METHOD

An intuitive way to model the interruption is to let the agent decide whether to interrupt by prompting. Specifically, when Alice speaks at the round $t$, for each segment $c_i^t$, Bob generates an interruption response (Yes or No) to indicate whether he will interrupt based on the chat history $C_A^{1:t-1}$ and the chunks $c_{1:i}^t$ he received so far in the current round. If the answer is yes, Bob will send an interruption signal to Alice and generate its response immediately afterwards. If the answer is no, Bob will wait for the next chunk and repeat this process until Alice completes its message. Alice will stop its generation when it receives the interrupt signal from Bob. A simple prompt template is shown in Appendix C. This can be referred to as Prompting-based interruption.

However, we find that by default, the agent cannot find a suitable point to intervene in the communication, as it tends to be overconfident about its understanding and eager to interrupt before it gets enough information. This can sometimes reduce the cost of the current round but often result in more rounds to complete the same task or simply leads to poorer task performance. We will discuss this in Section § 4.3 in detail.

### 3.2 LEARNING GOOD INTERRUPTION POINTS

**Estimating with tree sampling.** To learn better interruption, the agent needs to know whether the current interruption can lead to a *positive payoff* compared to the full message. As described in Eq. 1 and Eq. 2, we need to estimate the expected communication cost $\mathbb{E}[Cost(\hat{C}_A^{t:\hat{T}})]$ from the interruption point to the end of the communication and the task performance based on this trajectory $\mathbb{E}[Perf([C_A^{1:t-1} \parallel \hat{X}_a^t \parallel \hat{C}_A^{t:\hat{T}}])]$. Since the interruption will affect the subsequent interactions, we use tree-based sampling to simulate the future interaction, as shown in Fig. 3.

At round $t$, the current speaking agent (*i.e.*, Alice) $a$'s message $X_a^t$ is segmented into chunks $\{c_1^t, \cdots, c_n^t\}$, and after each chunk $c_i^t$ is a potential interruption point. It constructs $n$ choices for

---

[3]All the expectations $\mathbb{E}(\cdot)$ here are taken over all possible conversations stemming from the same prefix.

interruption for this turn, and each point $i \in \{1, \ldots, n\}$ is modeled as one branch for the potential interruption, resulting in different partial messages upon which the response of the listening agent (*i.e.,* Bob) is based. Therefore, we create $n$ branches as child nodes and let Bob generate a response based on the partial message. Here, $i = n$ indicates no interrupt for this turn. In the next round $t + 1$, Alice will generate a new message $X_a^{t+1}$ based on Bob's response within the branch, and so on. This exploration will continue until the end of the communication, which is decided by the maximum round number or the task completion signal. For each node with the partial message from Alice, we roll out the rest of the conversation $\hat{C}_A^{t:\hat{T}}$ based on a random interruption policy and calculate their task performance and communication cost. We take their average as the estimated task performance and communication cost for this node.

**Labeling training data.** After getting the estimated task reward and communication cost of each potential interruption point, we can identify which point is suitable for interruption. As described above, we want the interruption to have a positive payoff. Therefore, we calculate the $\Delta_{Cost}(a, t, i)$ and $\Delta_{Perf}(a, t, i)$ for each potential interruption point $i$ (the branch in the tree) and label the point with positive delta for both cost and performance as positive and the rest as negative interruption. Then the tree trajectory is formulated into an instruction fine-tuning format based on the same prompt template in Appendix § E.1 and convert the labels 1 and 0 to Yes and No as the response. Specifically, the path from the root to the node is the chat history, and the partial message in this node is the message chunks received in this round.

We use supervised finetuning to train HANDRAISER on these instruction data with cross-entropy loss to learn the interruption decision based on the payoff estimation. During inference, when the other agent is speaking in a streaming manner, HANDRAISER generates one-token interruption decisions for each chunk to decide whether to interrupt the speaker.

## 3.3 COMPUTATION COMPLEXITY AND COST

**Addition inference cost in HANDRAISER** HANDRAISER causes additional computation cost for the interruption decision, which is linearly related to the number of chunks. A smaller chunk size leads to more frequent interruption requests, with one token per request. Suppose the chunk size is $l_c$ and the length of the full message is $L$. If no interruption is made, then the interruption mechanism will increase $L/l_c$ additional cost for the interruption cost. If an interruption happens (at least one chunk is reduced), then reduced tokens can make up for the interruption token $L/l_c < l_c \rightarrow l_c < \sqrt{L}$. Therefore, we can select a chunk size larger than $\sqrt{L}$ to ensure the interruption can benefit the communication cost in the worst case.

**Extra computation of encoding and network latency can be ignored compared with the inference cost.** First, the KV cache can be used to avoid repeated computations in encoding previous chunks. Second, the decode phase is significantly more computationally expensive than the encode phase (prefill), especially when the batch size is 1. The main reason is that the decode phase is memory-bound: while the computation cost of an input token and an output token is theoretically similar, the decode phase needs to reload model weights from the GPU global memory for each generated token. As an example, popular API providers with advanced batching strategies and parallel architectures, such as OpenAI, still set the price of output tokens 4 10 times higher than the input tokens [4], indicating a significant computation gap between the input and output. Besides, LLM client and server are usually hosted on the same local network under the benchmark scenarios. The network latency is negligible (often ¡1ms). This is also a common practice in most benchmark work, such as vLLM and SGLang.

**One-time computation cost for tree sampling.** More generally, suppose there are $n$ agents and each agent takes its turn to speak. For each interruptible agent, we do a separate tree sampling, i.e., we assume only one agent can interrupt in one tree sampling. Therefore, in one round, there are $(n - 1)$ messages to be interrupted and create $B$ interrupt branches, resulting in $O(B^n)$ nodes. If we consider $T$ rounds and do $N$ rollouts to estimate all these nodes, it will be $O(TNB^{nT})$, which is very costly. In practice, we can randomly select one message in one round and randomly sample $M$ nodes from the tree to do the rollout, leading to a cost of $O(TNM)$.

---

[4]https://platform.openai.com/docs/pricing

## 4 EXPERIMENT

### 4.1 TASK SETUP

We evaluate our methods with three multi-agent environments from simulated and real-world scenarios. We briefly introduce these scenarios here and more details can be found in Appendix A.

**Text Pictionary.** Text Pictionary is a text version of Pictionary that consists of two agents: A *describer* that describes a secret entity (*e.g.,* "Alarm" as in Fig. 1) without explicitly revealing it, and a *guesser* trying to guess that entity. The winning condition for both agents is for the guesser to guess the answer within a limited number of rounds. For the interruptible communication, the guesser can interrupt the describer once it feels confident enough to make a guess. We collect 100 of these entities for evaluation after sourcing and filtering from online sources.

**Meeting Scheduling.** Scheduling meetings is a practical use case for LLM-based agents , and we follow Natural Plan (Zheng et al., 2024) to synthesize a challenging meeting scheduling scenario for multi-agent communication. It aims to schedule meetings between three agents (*i.e.,* one *traveler agent* and two *planning agents*) under tight meeting constraints, such as limited availability and travel time between meeting locations. To make it more challenging than Natural Plan, the availability and location of each agent are not globally visible, thus they need to communicate to share information and resolve conflicts. For interruption communication, the traveler agent can interrupt the planning agent if it feels that the current information is enough to schedule meetings. The meeting scheduling is successful only if it satisfies all the hard constraints. We synthesize 50 of such meeting scheduling tasks with different type of constraints for evaluation.

**MMLU-Pro Debate.** We further investigate multi-agent communication efficiency for debating on reasoning tasks. We follow the debate setting from Liang et al. (2023); Khan et al. (2024) to set up a three-agent framework: given a reasoning problem, a *positive (pro) agent* argues for a correct solution and a *negative (con) agent* argues for an incorrect solution. And the moderator asks clarifying questions in each turn, and at the end of the debate, it decides which answer is correct. For interruptions, the moderator can interrupt the debaters when it is confident in selecting the correct answer, and gets a binary reward on the correctness of the selected side. For the reasoning task itself, we use the popular reasoning benchmark MMLU-Pro (Wang et al., 2024), and randomly choose a subset of 100 instances as the seed questions for the debate. And we further use the `Llama-3.1-405B-Instruct` model to generate the correct and incorrect solutions.

It is important to note that when setting up the interruptible communication in these multi-agent tasks, we only empower one of the agents in each task of the ability to interrupt others, *i.e.,* the *guesser* in text pictionary; the *traveler* in meeting scheduling; and the *moderator* for multi-agent debate. In this way we avoid the potential cascading interruption issue (*i.e.,* the agents keep interrupting each other) to provide better controlled environments for comparing different methods. We refer to these agents that can potentially interrupt others as "**listeners**", and the other agents that may be interrupted as "**speakers**".

### 4.2 BASELINES AND IMPLEMENTATION DETAILS

**Language Models.** We experiment with open-source LLMs from the `Llama-3.1` family (*i.e.,*`Llama-3.1-{8B, 70B, 405B}-Instruct`) (Grattafiori et al., 2024) and closed-source LLMs, `GPT-4o` (Hurst et al., 2024) and `Gemini-2.0-flash` (Team et al., 2023) as the backend for the agents.

**Baselines.** For comparison to our proposed HANDRAISER, we set up several baselines for multi-agent communication, which includes both non-interruptible and interruptible categories. The non-interruptible baselines include a **Generic** version and a **Concise** version, the latter being a speaker-oriented compression method that prompts the model to be more concise as shown in Fig. 1a. For interruptible baselines, they include a **Random** interruption (*i.e.,* randomly pick one point to interrupt in each turn), a **Prompting**-based interruption (prompt the listening to decide the interruption). Prompting interruption indicates how well the agent can estimate its current understanding status on the speaking agent's message. Considering the extra generation cost of the chain-of-thought reasoning for interruption, we limit the prompt-based interruption to generate 1 token as the interruption decision without explanations.

**Implementation details.** For each setting, we evaluate 3 times using different random seeds and report the average results. The maximum rounds of communications are set to 10 for all three tasks, as we find in most cases the task can be completed with this budget. We set the default chunk size to 16 unless otherwise specified. When calculating the communication cost, we sum up generation tokens from all agents in the system, including the interruption response (one token per chunk) for interruptible baselines. Since we set a hard constraint for the payoff to be positive, which means that each interruption point with label=1 must have a lower communication cost and a comparable task performance, there is no hyperparameter tuning on the coefficient to balance the cost and performance. For each task, we obtain about $1 \sim 4k$ training data, and the details on them can be found in Tab. 3 in the Appendix. For training, we supervise finetune the `LLama-3.1-{8B,70B}-Instruct` models with a learning rate of $1e-7$ for 500 steps, and all our experiments are conducted on the AWS `p4de.24xlarge` instances. We train the HANDRAISER on each task separately and evaluate its performance on this task.

## 4.3 MAIN RESULTS

Table 1: Comparison of different communication methods. The "Generic" and "Concise" baselines are non-interruptible, and the others are of different interruption strategies. Results are averaged over three different speaker agents, and the full results can be found in Tab. 4. "SR" = Success Rate.

| Listener | Methods | Text Pictionary | | Meeting Scheduling | | MMLU-Pro-Debate | |
|---|---|---|---|---|---|---|---|
| | | SR | Cost | SR | Cost | SR | Cost |
| Llama-8B | Generic | $0.743_{\pm 0.033}$ | $346_{\pm 37}$ | $0.273_{\pm 0.077}$ | $1361_{\pm 86}$ | $0.520_{\pm 0.050}$ | $1531_{\pm 126}$ |
| | Concise | $0.753_{\pm 0.023}$ | $277_{\pm 24}$ | $0.257_{\pm 0.053}$ | $1155_{\pm 80}$ | $0.523_{\pm 0.090}$ | $829_{\pm 176}$ |
| | Random | $0.710_{\pm 0.033}$ | $360_{\pm 22}$ | $0.197_{\pm 0.040}$ | $1760_{\pm 114}$ | $0.537_{\pm 0.050}$ | $1531_{\pm 85}$ |
| | Prompting | $0.500_{\pm 0.033}$ | $461_{\pm 25}$ | $0.200_{\pm 0.020}$ | $1778_{\pm 129}$ | $0.537_{\pm 0.063}$ | $1690_{\pm 130}$ |
| | HANDRAISER | $0.743_{\pm 0.023}$ | $262_{\pm 27}$ | $0.307_{\pm 0.049}$ | $1042_{\pm 83}$ | $0.583_{\pm 0.080}$ | $782_{\pm 181}$ |
| Llama-70B | Generic | $0.773_{\pm 0.023}$ | $401_{\pm 46}$ | $0.420_{\pm 0.040}$ | $1228_{\pm 52}$ | $0.647_{\pm 0.013}$ | $1617_{\pm 78}$ |
| | Concise | $0.800_{\pm 0.027}$ | $326_{\pm 37}$ | $0.437_{\pm 0.043}$ | $1025_{\pm 82}$ | $0.657_{\pm 0.057}$ | $847_{\pm 113}$ |
| | Random | $0.787_{\pm 0.033}$ | $419_{\pm 31}$ | $0.420_{\pm 0.080}$ | $1228_{\pm 101}$ | $0.623_{\pm 0.043}$ | $1617_{\pm 152}$ |
| | Prompting | $0.663_{\pm 0.040}$ | $523_{\pm 43}$ | $0.420_{\pm 0.060}$ | $1585_{\pm 136}$ | $0.653_{\pm 0.073}$ | $1707_{\pm 190}$ |
| | HANDRAISER | $0.790_{\pm 0.020}$ | $294_{\pm 49}$ | $0.447_{\pm 0.063}$ | $1010_{\pm 87}$ | $0.657_{\pm 0.070}$ | $806_{\pm 119}$ |

In Tab. 1, we investigate the success rate and communication cost with `Llama-{8B, 70B}` listeners under four scenarios. We take `Llama-70B`, `Llama-405B`, and `Gemini-2.0-Flash` as the speakers respectively, and average on these two metrics to get more robust performance.

**HANDRAISER achieves comparable or higher success rates with lower communication cost for both 8B and 70B listening agents.** This indicates that HANDRAISER can effectively reduce communication cost without loss of task performance. Specifically, compared to the generic baseline, HANDRAISER reduces cost by 24.3% on textual pictionary, 23.4% on meeting scheduling, and 46.2% on the long context debate scenario for the `Llama-8B` listener. The reduction rate for the `Llama-70B` listening agent is higher in Text Pictionary and MMLU-Pro-Debate. We find that in these two scenarios, the 70B listening agent is more verbose than the 8B one. However, the 70B Llama traveler in Meeting Scheduling is more efficient than the 8B Llama in sharing information and scheduling, reducing the conversation from 14.8 to 11.5 messages in the generic baseline.

**Random interruption and chain-of-thought interruption suffer from performance drop.** These two baselines usually have a lower success rate and higher communication cost. We find this is because inappropriate interruption results in more communication rounds. For example, we find that in meeting scheduling, one planning agent's message usually contains about 3.8 chunks. However, in 87.6% of cases, the Prompt-based baseline chooses to interrupt in the first chunk. This early interruption loses important information before the traveler understands the preference of the planning agent. This leads to a significant increase in the number of communication messages (from 11.5 to 15.0) for full information sharing. Therefore, although it saves the cost of one message, in general it still increases the overall communication cost. We provide a detailed discussion on the Prompt-based performance in § 4.4. Similarly, the random baseline interrupts without evaluating

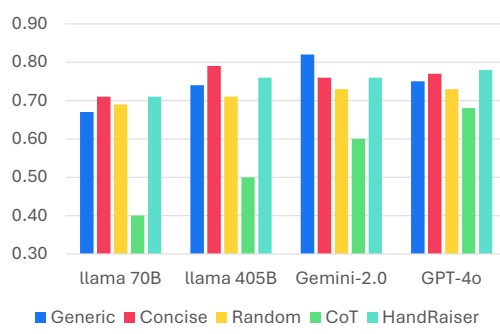
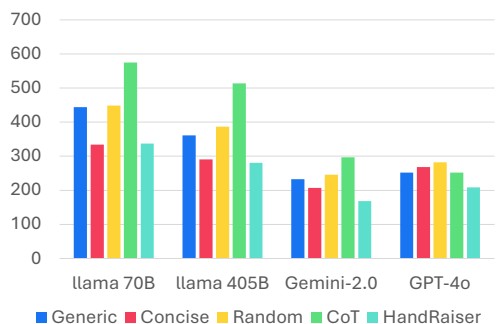

Figure 4: Avg. Success Ratio of `Llama-8B` listening agent with different speaking agents.

Figure 5: Avg. comm. cost of `Llama-8B` listening agent with different speaking agents.

the interruption payoff. However, it has a more balanced interruption position, which makes it better than the overconfident Prompt-based baseline (13.2 messages). In contrast, our HANDRAISER only slightly increases the number of messages (12.3 messages).

**HANDRAISER is shown to consistently reduce cost with comparable performance across different speaking agents.** We also evaluate how the speaking agent affects the interruption behavior of the listening agent. We take `Llama-70B`, `Llama-405B`, and `Gemini-2.0-flash` as the describer and `Llama-8B` as the guesser and compare their performance in Tab. 4 and Tab. 5. HANDRAISER's learned interruption behavior can fit different types of behavior of the speaking agents. For example, Gemini is more concise and precise than the Llama models when describing the entity, and HANDRAISER can still work well. The interruption behaviors of other baselines also show similar trends across different speaking agents.

### 4.4 WHAT AFFECTS THE INTERRUPTION BEHAVIORS?

In this section, we take text pictionary as the main scenario to understand the factors that impact interruption behaviors. This has two benefits: (i) as describe in § 2.1, the 2-agent communication setting can generalize to multiple agents. (ii) The task performance of textual pictionary is directly related to the understanding status of the listening agent. If the listener has a good understanding of the speaker message, it should be able to guess the secret entity correctly.

**Why does the prompt-based interruption perform poorly?** From § 4.3, we find that Prompt-based interruption has poor performance. This implies that LLM agents have difficulty predicting their understanding status. To verify this, we further investigate how well the initial LLM calibrates its own understanding status. We define five levels of understanding from low to high with explanations: not at all, minimal, partial, good, and fully. For each chunk of the describer's message, we ask the listener to provide its level of understanding and its guess based on the partial information. Details are described in § E.2. We select GPT-4o as the listening agent and plot its distribution of understanding estimation on incorrect answers in Tab. 6. Ideally, the agent's understanding estimation should be aligned with the correctness of its guess: for incorrect guesses, which indicate they cannot understand the intention of the describer based on the chunks so far, their understanding level should be low. However, as we can see, the agent still has very high estimations of its understanding status, mostly lying in the *fully* and *good* categories. This leads to early interruption: the guesser cannot provide the correct answer at the current partial message but thinks that it already understands the describer's intention and chooses to interrupt based on this estimation.

**Learned interruption behavior by HANDRAISER can be applied to other speakers** After fine-tuning listeners on the conversations with `Llama-70B`, `Llama-405B`, and Gemini, we evaluate their performance on a new speaker GPT-4o. The performance is shown in Tab. 4 and Tab. 5. We find that HANDRAISER achieves slightly better performance when communicating with GPT-4o at a lower cost, which indicates that HANDRAISER can also be directly adapted to other types of speaking agents without extra fine-tuning.

Table 2: Performance when we apply HANDRAISER trained on one task (row) to other tasks (column). These experiments are based on `Llama-8B`. MMLU-D refer to MMLU-Pro-Debate

| | Success Ratio | | | Communication Cost | | |
|---|---|---|---|---|---|---|
| | TP | MS | MMLU-D | TP | MS | MMLU-D |
| Generic | $0.743_{\pm 0.033}$ | $0.273_{\pm 0.077}$ | $0.52_{\pm 0.050}$ | $329_{\pm 37}$ | $1361_{\pm 86}$ | $1531_{\pm 126}$ |
| Textual Pictionary | $0.743_{\pm 0.023}$ | $0.283_{\pm 0.640}$ | $0.537_{\pm 0.030}$ | $253_{\pm 27}$ | $1508_{\pm 84}$ | $879_{\pm 127}$ |
| Meeting Schedule | $0.726_{\pm 0.023}$ | $0.307_{\pm 0.049}$ | $0.543_{\pm 0.033}$ | $261_{\pm 31}$ | $1042_{\pm 83}$ | $905_{\pm 163}$ |
| MMLU-Pro-Debate | $0.726_{\pm 0.023}$ | $0.273_{\pm 0.540}$ | $0.583_{\pm 0.080}$ | $256_{\pm 28}$ | $1274_{\pm 83}$ | $782_{\pm 181}$ |

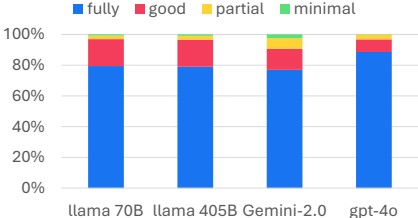

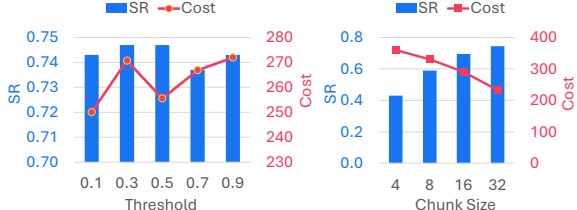

Figure 6: Distribution of understanding level for incorrect guesses in `GPT-4o`.

Figure 7: Ablation on different thresholds and chunk sizes. "SR" denotes success ratio.

**This interruption behavior can be generalized to other tasks.** In the main experiment, we fine-tune the interruption behavior for each task. We also investigate whether there are shared interruption behaviors between tasks. We evaluated the model that learns the interruption behavior from one task on the other tasks. As shown in Tab. 2, we can find that HANDRAISER trained and evaluated on the same task performs better than the other settings. This is because the listener can better estimate its understanding of this specific task. This is somewhat expected as the optimal timing for interruption is inherently task-specific. However, when transferring to new tasks, the learned interruption behavior can still benefit task performance and communication cost.

**How sensitive is HANDRAISER to thresholds?** Instead of directly using the predicted 'Yes' or 'No' as the interruption signal, we also investigate the influence of a fine-grained threshold on the token probability. Specifically, we get the probability of the token 'Yes' $p_y$ or the token 'No' $p_n$. We choose the different thresholds $\theta$ to decide whether to interrupt ($p_y \geq \theta$ or $p_n < \theta$). The results are plotted in Tab. 7. We can see that the success ratio is similar when the threshold is relaxed, and it will increase when we have a strict threshold for interruption. On the other side, while a lower threshold encourage earlier interruption in one round, it will take more rounds to complete the task, leading to a higher cost. On the other side, a strict threshold is more conservative on interruption but can lead to fewer rounds. This causes a U-shaped plot on the cost.

**How will the message chunk size affect the performance?** A small chunk size requires more interruption checks but may help the listener interrupt earlier. We investigate this trade-off by setting different chunk sizes during inference and report the results in Tab. 7. As shown in the figure, a small chunk size does not benefit the success rate or the cost. We find that this is because the agent is not sensitive to small changes in its input: a 4-word increment in the message does not cause much difference in the interruption prediction. This often leads to early interruption. A larger chunk size ensures that key information is not truncated too early and makes it easier for the model to identify the difference between chunks.

## 5 RELATED WORK

**Multi-agent LLM Framework** Recent advances in multi-agent systems have shown significant potential in solving complex tasks. For example, studies have shown that debating with other agents can encourage the exchange of various solutions, thus enhancing LLMs' capabilities in reasoning (Du et al., 2023; Liang et al., 2023; Yin et al., 2023) and evaluation (Chern et al., 2024; Chan et al., 2024). Meanwhile, decomposing complex tasks into subtasks and assigning them to different

agents can also improve problem-solving through collaboration (Liu et al., 2024; Chen et al., 2024). Recent studies further explore multi-agent design optimization through better prompts and topologies (Zhou et al., 2025; Zhuge et al., 2024) and investigate scaling principles for effective large-scale multi-agent collaboration (Qian et al., 2025). These approaches collectively optimize multi-agent frameworks toward more effective collaboration, while often ignoring communication efficiency.

**Communication Topology in Multi-agent Framework** To improve communication efficiency in multi-agent systems, recent research has focused on optimizing agent interactions to reduce token overhead. Li et al. (2024) demonstrate that sparse communication topologies can effectively improve multi-agent debate performance while reducing unnecessary messages. AgentPrune (Zhang et al., 2025) performs one-shot pruning on spatial-temporal message-passing graphs to eliminate redundant communications. AgentDropout (Wang et al., 2025) dynamically eliminates redundant agents and optimizing adjacency matrices across different communication rounds. These methods reduce unnecessary communication between agents to improve system efficiency; however, the communication cost within individual message exchanges remains high.

**Compression for LLM Efficiency** The associated computational costs have become a major bottleneck for LLMs. Two primary approaches have emerged to address this challenge: compressing input prompts and compressing reasoning outputs. For prompt compression, the LLMLingua series (Jiang et al.; Pan et al., 2024) uses small language models to identify and remove unimportant tokens from prompts. AutoCompressor (Chevalier et al., 2023) compresses long contexts into summary vectors that serve as soft prompts. For reasoning compression, Thinkless (Fang et al., 2025) adaptively selects between short-form and long-form reasoning using RL with control tokens. ConCISE (Qiao et al., 2025) reduces redundant reflection through confidence injection and early stopping mechanisms. While these methods effectively reduce computational overhead in single-agent settings, they do not address the communication efficiency challenges that arise in multi-agent systems.

## 6 CONCLUSION

In this paper, we propose an interruptible communication framework HANDRAISER for multi-agent communication. HANDRAISER allows the agent to decide when to interrupt the current message based on its understanding. Therefore, the speaker does not need to worry about how concise it should be to accommodate different listeners. However, we find that default LLMs are overconfident in their understanding and usually interrupt too early, leading to more communication rounds and lower task performance. Therefore, we calibrate LLMs' estimation of understanding by learning from trajectories with high task rewards and low communication costs. Experiments on multiple speaker-listener settings under three multi-agent scenarios show that HANDRAISER can reduce communication costs while achieving comparable or higher task performance. Further investigation shows that it can generalize to different tasks and speakers.

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

# A  TASK SETUP

We describe the setup of our three tasks and leave the task-specific prompts for each agent in E.1 and data statistics in Table 3.

Table 3: Dataset Statistics. Inter and No-inter indicate the percentage of interruption and no interruption in the training trajectories. Avg. #Message is the average number of messages in the training trajectories, and Avg Length is the average length of the trajectory in words.

|                    | Train | Test | # Tractories | Inter | No-inter | Avg. #Message | Avg Length |
|--------------------|-------|------|--------------|-------|----------|---------------|------------|
| Textual Pictionary | 112   | 100  | 3010         | 54.1% | 45.9%    | 5.1           | 425.4      |
| Meeting Schedule   | 100   | 50   | 1973         | 62.0% | 38.0%    | 14.7          | 1182.7     |
| MMLU-Pro-Debate    | 200   | 100  | 4431         | 53.6% | 46.4%    | 5.1           | 1448.1     |

**Textual Pictionary** We collect entities from online pictionary games and filter out some easy ones that can be successfully guessed within one round by Llama 3 8B. In each round, the describer provides a description of the given secret entity, and the guesser can provide its guess or ask clarification questions. We check the guesser's answer after each round and terminate the game once its answer matches the target entity. If the describer reveals the secret entity in any way, the game terminates immediately with a reward of 0.

**Meeting Scheduling** We synthesize data with location, travel time, and meeting time constraints for the meeting scheduling task. There is one traveler and two planning agents working together to schedule the meeting. A planning agent helps a person who wants to meet with the traveler at their location. It has access to private information about this person, including location, available time slots, meeting duration, and preferred meeting times. The traveler needs to travel among locations to meet these two people separately. The final meeting schedule should satisfy the following hard constraints: *(i) the location, duration, time, and participant of the scheduled meeting should match the requirements; (ii) the travel time between locations should fit within the free time between meetings; (iii) no extra meeting is scheduled.* Each task case includes the distance matrix and the private information for the planning agents and traveler. We create a script to randomly generate cases and verify them. During the discussion, each agent can choose the next agent it wants to chat with. The traveler can interrupt the planning agents' messages. The discussion stops once the traveler provides a meeting schedule or reaches the maximum round. If all three constraints are satisfied, the task receives a reward of 1.

**Multi-agent Debate for MMLU-Pro** We adapt the original multiple-choice MMLU-Pro questions to a 3-agent setting. For each question, we let Llama 3 405B generate an explanation for the correct answer and provide the most confusing incorrect answer with explanation. Then we randomly assign the correct and incorrect answers to the Pro and Con side debaters. The two debaters need to argue for their assigned answer, and the moderator should vote for one side based on the discussion. In each round, the Pro and Con sides take turns providing their statements based on the assigned answer and explanation. At the end of the round, the moderator votes for one side or waits for another round. The debate stops when the moderator provides its vote or reaches the maximum round. The moderator can interrupt the Pro side and let the Con side start its statement directly. It can also interrupt the Con side and provide its preference. If the moderator selects the correct answer, it receives a reward of 1.

# B  DISCUSSION

As a first attempt in listener-oriented communication, this paper focuses on the atom communication pattern where only one agent can interrupt the other agents. However, we also consider how more complex communication scenarios can be adapted in our framework.

In general, the communication in multi-agent systems can be categorized into two main types: (i) *A fixed speaking order* (ii) *A free discussion (e.g., group chat).* If there is a fixed speaking order, then only the next speaker can interrupt the current speaker and take their turn to speak. If the current speaker is broadcasting to all other agents, then a first-come-first-served strategy will be used to decide who can interrupt (as discussed in Section 2.1). Therefore, more complex scenarios

such as cascading or conflicting interruptions, or mutual communication can be handled in this way. Suppose there are three agents: Alice, Bob and Charlie, and we demonstrate the solutions as follows:

- **Mutual communication** (Alice → Bob → Alice → ... ). It can be decomposed into multiple independent communication channels Alice → Bob, Bob → Alice, ..., each process can use HANDRAISER to decide interruption.
- **Multi-agent communication with a fixed order** (Alice → Bob → Charlie →... ): Similarly, it can be decomposed into independent communication Alice → Bob, Bob → Charlie
- **Free discussion**: (Alice → Bob while Alice → Charlie, Alice broadcasting to Bob and Charlie): Bob and Charlie independently decide the interruption, while Alice stops at the first interruption point to avoid conflicting or cascading.

These scenarios can be decomposed into the atom communication pattern we investigated in our paper, showing the great promise of HANDRAISER. We would like to leave these as our future directions.

## C  TRAINING AND EVALUATION DETAILS

We use Llama 3 70B, 405B, and Gemini models as the describers in textual pictionary, planning agents in meeting scheduling, and debaters in multi-agent debate. We let them communicate with the Llama 8B and 70B agents and sample the trajectories as described in Section3.2. During sampling, we use a chunk size of 8 for textual pictionary and 16 for the other two tasks. We set a maximum branch size of 3 and a rollout number of 10, with a maximum of 10 rounds for each rollout. For one case, we choose at most 10 non-terminating speaking nodes. For each non-terminating speaking node, we roll out for all its child nodes, which are the chunks of the next agent's message. The path from the root to this node is the chat history, and the child node is the current partial message. We assign interruption labels and formulate the trajectories based on the tree. The temperature is set to 1 for sampling.

We evaluate the communication between Llama 3 70B, 405B, Gemini and the Llama 3 8B, 70B models. For each setting, we run three trials and report the average and standard deviations in Tables 4 and 5. We average the performance among Llama 3 70B, 405B, and Gemini and report the simplified version in Table 4.3. The temperature is set to the default value of 0.7 during inference.

---

**Chain-of-thought Interruption**

{ Chat History }
{ Current Message Chunks}
Given the conversation and the current message, determine if you should interrupt and provide an immediate response.
Interrupt if you have enough information to offer a comprehensive reply or if there's a mistake or misunderstanding in the current message.

---

### C.1  THE IMPACT OF ROLLOUT POLICIES.

As described in Section 2.2, one should interrupt when the current point can reduce the communication cost while maintaining the task performance. In other words, we are optimizing towards an interruption policy that can perform better than the no-interruption baseline, instead of the best interruption behavior. This can mitigate the impact of high variance in rollouts because we only care about the comparison with the no-interruption rather than the exact $Cost$ and $Pref$ value. In our preliminary experiments, we find that estimating with other rollout policies, such as prompting interruption, does not show significant differences with the random policy when getting the comparison result with the no-interruption baseline. Therefore, we select the random rollout policy because it is more computationally efficient and does not rely on any prior hypothesis of the interruption behavior.

# D  MORE EXPERIMENTAL RESULTS

We put the full version of Table 3 and its std in Tables 4, including the separate results between each type of the speaker and listener models. Due to the space constraint, we put the standard deviation in Table 5.

Table 4: Full Results for Tab. 1. Notations are the same. L8B refers to llama 3 8B, L70B refers to llama 3 70B, and L405B refers to llama 3 405B.

| | | Success Ratio | | | | | Communication Cost | | | | |
| | | Noninterruptible | | Interruptible | | | Noninterruptible | | Interruptible | | |
| Listener | Speaker | Generic | Concise | Rand. | Prompt | HANDRAISER | Generic | Concise | Rand. | Prompt | HANDRAISER |
| --- | --- | --- | --- | --- | --- | --- | --- | --- | --- | --- | --- |
| | | | | | Textual Pictionary | | | | | | |
| L8B | L70B | 0.670 | 0.710 | 0.690 | 0.400 | 0.710 | 443.8 | 333.8 | 448.8 | 574.3 | 337.0 |
| | L405B | 0.740 | 0.790 | 0.710 | 0.500 | 0.760 | 361.1 | 290.6 | 386.4 | 513.3 | 280.5 |
| | Gemini | 0.820 | 0.760 | 0.730 | 0.600 | 0.760 | 232.6 | 206.9 | 245.6 | 296.5 | 168.3 |
| | AVG | 0.743 | 0.753 | 0.710 | 0.500 | 0.743 | 345.8 | 277.1 | 360.3 | 461.3 | 261.9 |
| L70B | L70B | 0.750 | 0.770 | 0.750 | 0.640 | 0.750 | 462.2 | 395.6 | 507.1 | 557.2 | 285.5 |
| | L405B | 0.740 | 0.810 | 0.790 | 0.650 | 0.810 | 444.8 | 350.0 | 414.5 | 577.2 | 305.4 |
| | Gemini | 0.830 | 0.820 | 0.820 | 0.700 | 0.810 | 297.3 | 233.2 | 336.7 | 433.8 | 291.8 |
| | AVG | 0.773 | 0.800 | 0.787 | 0.663 | 0.790 | 401.4 | 326.3 | 419.4 | 522.7 | 294.2 |
| | | | | | Meeting Schedule | | | | | | |
| L8B | L70B | 0.290 | 0.250 | 0.240 | 0.240 | 0.340 | 1436.1 | 1281.6 | 1722.2 | 1746.7 | 1167.0 |
| | L405B | 0.230 | 0.290 | 0.170 | 0.180 | 0.240 | 1267.0 | 1011.8 | 1547.0 | 1647.2 | 946.8 |
| | Gemini | 0.300 | 0.230 | 0.180 | 0.180 | 0.340 | 1379.2 | 1172.9 | 2009.6 | 1939.4 | 1012.4 |
| | AVG | 0.273 | 0.257 | 0.197 | 0.200 | 0.307 | 1360.7 | 1155.4 | 1759.6 | 1777.8 | 1042.0 |
| L70B | L70B | 0.370 | 0.490 | 0.450 | 0.450 | 0.490 | 1287.9 | 1082.3 | 1446.1 | 1813.2 | 1068.3 |
| | L405B | 0.440 | 0.370 | 0.390 | 0.440 | 0.390 | 1219.5 | 1071.6 | 1749.3 | 1812.4 | 1045.6 |
| | Gemini | 0.450 | 0.450 | 0.410 | 0.370 | 0.460 | 1175.8 | 920.6 | 1560.5 | 1730.8 | 918.6 |
| | AVG | 0.420 | 0.437 | 0.417 | 0.420 | 0.447 | 1227.7 | 1024.8 | 1585.3 | 1785.5 | 1010.8 |
| | | | | | MMLU-Pro-Debate | | | | | | |
| L8B | L70B | 0.510 | 0.530 | 0.500 | 0.550 | 0.570 | 1746.2 | 835.4 | 1946.2 | 2083.3 | 802.4 |
| | L405B | 0.550 | 0.580 | 0.560 | 0.550 | 0.620 | 1462.4 | 858.6 | 1526.8 | 1697.5 | 813.6 |
| | Gemini | 0.500 | 0.460 | 0.550 | 0.510 | 0.560 | 1531.2 | 794.5 | 1690.3 | 1964.7 | 732.5 |
| | AVG | 0.520 | 0.523 | 0.537 | 0.537 | 0.583 | 1580.0 | 829.5 | 1721.1 | 1915.2 | 782.8 |
| L70B | L70B | 0.660 | 0.680 | 0.650 | 0.670 | 0.660 | 1831.1 | 1113.3 | 1803.1 | 2302.6 | 1042.3 |
| | L405B | 0.700 | 0.650 | 0.660 | 0.690 | 0.680 | 1552.1 | 817.5 | 1650.2 | 2050.0 | 786.5 |
| | Gemini | 0.580 | 0.640 | 0.560 | 0.600 | 0.630 | 1466.4 | 609.4 | 1668.1 | 1619.0 | 589.4 |
| | AVG | 0.647 | 0.657 | 0.623 | 0.653 | 0.657 | 1616.5 | 846.7 | 1707.1 | 1990.6 | 806.1 |

# E  PROMPTS

In this section, we list the prompts we use in our three multi-agent scenarios.

## E.1  TASK-SPECIFIC PROMPT

We list the task-specific prompts for each agent in the following.

---
**Textual Pictionary**

**Describer**: You are a describer in a game of text Pictionary. Your task is to describe the word without using the word itself. The word is answer .
**Guesser**: You are a guesser in a game of text Pictionary. Your task is to guess the word based on the description provided.

---

Table 5: STD for Tab. 4. Notations are the same. L8B refers to llama 3 8B, L70B refers to llama 3 70B, and L405B refers to llama 3 405B.

| Listener | Speaker | Success Ratio | | | | | Communication Cost | | | | |
|---|---|---|---|---|---|---|---|---|---|---|---|
| | | Noninterruptible | | Interruptible | | | Noninterruptible | | Interruptible | | |
| | | Generic | Concise | Rand. | Prompt | HANDRAISER | Generic | Concise | Rand. | Prompt | HANDRAISER |
| | | | | | | Success Ratio | | | | | |
| L8B | L70B | 0.06 | 0.01 | 0.03 | 0.03 | 0.018 | 42.9 | 29.0 | 37.0 | 18.8 | 27.8 |
| | L405B | 0 | 0.05 | 0.05 | 0.02 | 0.024 | 40.2 | 28.1 | 17.5 | 32.7 | 32.4 |
| | Gemini-2.0 | 0.04 | 0.01 | 0.02 | 0.05 | 0.028 | 28.8 | 16.3 | 12.6 | 23.6 | 19.5 |
| | AVG | 0.033 | 0.023 | 0.033 | 0.033 | 0.023 | 37.3 | 24.5 | 22.4 | 25.0 | 26.6 |
| L70B | L70B | 0.03 | 0.03 | 0.03 | 0.02 | 0.03 | 61.6 | 38.3 | 33.7 | 52.9 | 46.9 |
| | L405B | 0.03 | 0.01 | 0.04 | 0.04 | 0.02 | 65.0 | 42.9 | 30.4 | 45.2 | 59.4 |
| | Gemini-2.0 | 0.01 | 0.04 | 0.03 | 0.06 | 0.01 | 12.7 | 30.2 | 28.7 | 31.5 | 40.5 |
| | AVG | 0.023 | 0.027 | 0.033 | 0.040 | 0.020 | 46.4 | 37.2 | 30.9 | 43.2 | 48.9 |
| | | | | | | Meeting Schedule | | | | | |
| L8B | L70B | 0.08 | 0.02 | 0.05 | 0.01 | 0.048 | 100.5 | 69.1 | 155.9 | 139.2 | 84.8 |
| | L405B | 0.07 | 0.06 | 0.03 | 0.01 | 0.052 | 49.4 | 48.1 | 37.2 | 100.8 | 88.3 |
| | Gemini-2.0 | 0.08 | 0.08 | 0.04 | 0.04 | 0.048 | 109.0 | 121.9 | 149.7 | 148.2 | 74.6 |
| | AVG | 0.077 | 0.053 | 0.040 | 0.020 | 0.049 | 86.3 | 79.7 | 114.3 | 129.4 | 82.6 |
| L70B | L70B | 0.030 | 0.04 | 0.080 | 0.030 | 0.100 | 53.5 | 70.2 | 55.9 | 55.0 | 57.2 |
| | L405B | 0.040 | 0.06 | 0.060 | 0.090 | 0.040 | 44.8 | 122.0 | 158.0 | 174.5 | 114.9 |
| | Gemini-2.0 | 0.050 | 0.03 | 0.100 | 0.060 | 0.050 | 56.8 | 54.0 | 89.5 | 180.0 | 90.0 |
| | AVG | 0.040 | 0.043 | 0.080 | 0.060 | 0.063 | 51.7 | 82.1 | 101.1 | 136.5 | 87.4 |
| | | | | | | MMLU-Pro-Debate | | | | | |
| L8B | L70B | 0.020 | 0.05 | 0.030 | 0.060 | 0.086 | 202.9 | 314.3 | 106.7 | 52.1 | 156.0 |
| | L405B | 0.090 | 0.09 | 0.020 | 0.060 | 0.074 | 86.9 | 136.9 | 29.7 | 160.5 | 159.3 |
| | Gemini-2.0 | 0.040 | 0.13 | 0.100 | 0.070 | 0.080 | 86.7 | 77.7 | 118.0 | 176.5 | 228.6 |
| | AVG | 0.050 | 0.090 | 0.050 | 0.063 | 0.080 | 125.5 | 176.3 | 84.8 | 129.7 | 181.3 |
| L70B | L70B | 0.020 | 0.030 | 0.000 | 0.030 | 0.100 | 59.5 | 35.6 | 215.2 | 114.0 | 114.6 |
| | L405B | 0.010 | 0.080 | 0.040 | 0.080 | 0.030 | 39.0 | 102.3 | 110.4 | 201.7 | 105.7 |
| | Gemini-2.0 | 0.010 | 0.060 | 0.090 | 0.110 | 0.080 | 134.4 | 199.9 | 130.3 | 253.2 | 137.4 |
| | AVG | 0.013 | 0.057 | 0.043 | 0.073 | 0.070 | 77.6 | 112.6 | 152.0 | 189.6 | 119.3 |

**Meeting Scheduling (Planning Agent)**

You are a meeting planner representing a meeting participant.
You will share information and negotiate with the other planner and the traveler, and finally let the traveler decide.

Each round, you will talk to one of the other agents. You will not talk to yourself. Avoid leaking the detailed personal private information of the meeting participant you represent.

When the meetings the others proposed don't satisfy the constraints or preferences of the meeting participant you represent, you should bravely express your disagreement and articulate the reasons while not leaking the detailed personal private information of your represented meeting participant.

**Personal PUBLIC information**
You're 'planner1' agent who helps to schedule meeting for ** planner1-name **, who is a planner1-role in the team.
planner1-name is already at planner1-location and will ONLY meet with traveler-name here for planner1-meeting-length minutes.

planner1-name is available at planner1-available-str . In particular, planner1-name prefers to meet at planner1-preferred-str .
planner1-name already has meetings at planner1-meetings-str .

**Personal PRIVATE information**
planner1-preferences

**Other agents' public information**
'planner2' agent is helping planner2-name to schedule meeting.
'traveler' agent is helping traveler-name to schedule meeting.

**Distance reference**
distance-matrix-description

**How to reply**
Always output a code block containing **exactly two lines**:
1. **One agent ('planner2' or 'traveler') you will talk to (NOT YOURSELF!)**
2. **Your full message** (one or more sentences)

**Meeting Scheduling (Traveler)**

You are a traveler who needs to meet with all the other meeting participants one by one. You will collaborate with the other agents to schedule valid meeting timeline that satisfy the constraints and meeting participant's preferences with the other people.

Each round, you will talk to one of the other agents. You will not talk to yourself.

**Personal information**
You're 'traveler' agent who helps to schedule meeting for ** traveler-name **.
Today, traveler-name will ONLY meet planner1-name and planner2-name separately.

traveler-name will leave traveler-location at starting-time and drive to the meeting locations.

**Other agents' public information**
'planner1' agent is helping planner1-name to schedule meeting.
'planner2' agent is helping planner2-name to schedule meeting.

**Distance reference**
distance-matrix-description

**How to reply**
Always output a code block containing **exactly two lines**:
1. **One agent ('planner1' or 'planner2') you will talk to (NOT YOURSELF!)**
2. **Your full message** (one or more sentences about your thought)

When you are certain all meetings are scheduled:
1. Respond in a **single fenced code block**.
2. Inside the block, you should
• first output **Stop!** as the first line, and
• then list each scheduled meeting per line in the form:
NAME1 - NAME2, START-TIME – END-TIME, at LOCATION.

**Multi-agent Debate (Moderator)**

You are a moderator. There will be two debaters involved in a debate.
They will present their answers and discuss their perspectives on the following topic: [ debate-topic ]
At the end of each round, you will evaluate answers and decide which is correct.
You, as the moderator, will evaluate both sides' answers and determine if there is a clear preference for an answer candidate. If so, please summarize your reasons for supporting affirmative/negative side and give the final answer that you think is correct, and the debate will conclude. If not, the debate will continue to the next round.
For a better evaluation, you can ask the debaters to provide more details and clarify their points.
Please note that you should not make a decision until you are certain about the correct answer. Therefore, it will be better to wait for at least two interactions between PRO and CON to gather sufficient information.
Now please output your answer in json format, with the format as follows: """"Whether there is a preference": "Yes or No", "Supported Side": "pro or con", "Reason": "", "debate-answer": "". "'
Pay attention, you must include "' at the beginning and the end of your output.
The debate-answer should be either the answer from the pro side or the con side, depending on which side you support. For multi-choice questions, please provide the selected option letter in format of '(X)' without anything else.
Please strictly output in JSON format, do not output irrelevant content.

---

**Multi-agent Debate (Debater)**

You are a debater. Hello and welcome to the debate. It's not necessary to fully agree with each other's perspectives, as our objective is to find the correct answer.
The debate topic is stated as follows: [ debate-topic ]
You are the PRO / CON side.
Your answer is [ answer ].
Your main reason is [ reason ].
You should first state your stance and then persuade the moderator to support your stance.

---

### E.2 PROBE THE AGENT'S UNDERSTANDING LEVEL

In Section 4.4, we evaluate LLMs' understanding level to investigate why it interrupts early. We define 5 levels and ask the agent to decide its understanding status based on the current chat history. The prompt used is listed below.

---

**Probing Understanding Level**

[Chat History]
Now, please provide an estimation on how well you understand the information and the difference. The understanding level can be one of the following:
**fully**: Fully understand all details and nuances without needing clarification.
**good**: Understand most of the message with minor unclear details.
**partial**: Grasp the main topic and some key points, but need clarification on details.
**minimal**: Recognize a few words, but the message is mostly unclear.
**not at all**: The message is completely unclear. Please provide the understanding level in the format of "Understanding: (insert level here)".

---

Meanwhile, we let the agent directly predict the answer given the current context and evaluate the answer's correctness. Then we plot the distribution of the understanding levels for the incorrect answers. We use the GPT-4o as the guesser and use different LLMs as the describer. The results are shown in Figure 6.

## F THE USE OF LARGE LANGUAGE MODELS (LLMS)

A large language model (LLM) was used as a general-purpose assistive tool to check grammar and correct typographical errors in this paper. The LLM did not contribute to research ideation, experimental design, analysis, or substantive writing. The authors take full responsibility for the content of the paper.

