# OpenReview forum: "Learning to Interrupt in Language-based Multi-agent Communication"
_ICLR.cc/2026/Conference — Submitted to ICLR 2026_

### Official Review · Reviewer_w2uu · 2025-10-20

**Soundness:** 3
**Presentation:** 3
**Contribution:** 3
**Rating:** 4
**Confidence:** 4

**Summary:**

This paper proposes HANDRAISER, an interruptible communication framework for large language model (LLM)-based multi-agent systems. Unlike previous speaker-oriented compression approaches, it introduces a listener-oriented mechanism that allows agents to interrupt speakers dynamically during message generation to reduce redundant communication. The method combines a chunk-based streaming protocol with a learning strategy that predicts optimal interruption points based on expected task rewards and communication costs. Experiments across three scenarios—textual Pictionary, meeting scheduling, and multi-agent debate—show that HANDRAISER significantly reduces communication cost while maintaining comparable task performance.

**Strengths:**

1.This paper tackles a valuable and interesting problem. To address the issues of communication redundancy, context overload, and high reasoning cost in current multi-agent systems, this paper introduce a listener-oriented optimization framework. This perspective is distinct from the conventional speaker-oriented compression paradigm and is conceptually insightful.

2. The paper further proposes a simple but efficient method to realize interruptible communication.

**Weaknesses:**

1. My main concern is that the evaluation environments appear task-specific rather than representative of a general, open multi-agent system.
Across the three evaluated tasks, the interrupter’s role is more akin to a “user” or “supervisor” rather than an autonomous agent within the system. This one-directional communication topology makes the experiments resemble human–agent interaction more than multi-agent collaboration, where dialogue participants have symmetric communication rights.
Therefore, the reported results mainly demonstrate the effectiveness of a single-listener interruption mechanism, rather than that of a fully scalable multi-agent communication framework.
2. The framework lacks validation in general multi-agent settings that allow multi-directional interruption, where agents can act as both speakers and listeners. Its stability and scalability in open environments (e.g., GAIA, AIME) therefore remain uncertain. As a result, the paper reads more like a conceptual proposal introducing an interesting idea, but lacks sufficient empirical evidence to demonstrate its effectiveness in general multi-agent systems.

**Questions:**

See Weakness.

---

> ### Author Response · Authors · 2025-11-24
>
> **W1: My main concern is that the evaluation environments appear task-specific rather than representative of a general, open multi-agent system. This one-directional communication topology makes the experiments resemble human–agent interaction more than multi-agent collaboration, where dialogue participants have symmetric communication rights.**
>
> A1: We thank the reviewer for raising the important discussion about how our method can be applied to a general, open multi-agent system. As a first attempt in listener-oriented communication, this paper focuses on the atom communication pattern where only one agent can interrupt the other agents. However, the symmetric communication or more complex scenarios can also be adapted to our framework by:
>
> - **Symmetric communication can be decomposed into two one-directional communications in adjective time.** When we have Alice→Bob and Bob→Alice, it can be viewed as Alice→Bob at time $t$ and Bob→Alice at time $t+1$. In our paper, we focus on a single communication process and assume only Bob can interrupt. The interruption at time $t$ for Alice is similar.
> - **When the system is scaled up with more agents, their communication can be decomposed into multiple simultaneous communications at the same time.** When scaled up from 2 agents to 3 agents (Alice, Bob, and Charlie), there are at most 3 simultaneous communication processes (Alice→Bob, Alice→Charlie, and Bob→Charlie), and we can apply our HandRaiser to each single communication. Similar for more agents.
>
> To conclude, the communication in scalable Multi-agent systems can be complex, but they can be decomposed into multiple atomic communication processes (Alice→Bob) at the same or different time steps. Our HandRaiser can be applied to the atomic communication process to save communication costs, which shows the promise of our work. We would like to leave these promising directions as future work to work on. We have added this discussion in the **Appendix B Discussion** of our revised manuscript.
>
> ---
>
> **W2: The framework lacks validation in general multi-agent settings that allow multi-directional interruption, where agents can act as both speakers and listeners. Its stability and scalability in open environments (e.g., GAIA, AIME), therefore, remain uncertain.**
>
> A2: In this paper, we not only evaluate our framework on reasoning tasks (*Multi-agent Debate for MMLU-Pro*), which is the most popular scenario for communication in multi-agent systems [1,2,3,4], but also explore other scenarios such as *multi-agent games[5,6] (Text Pictionary) and planning with non-unique answers [7,8] (Meeting Scheduling).* The tasks, such as GAIA and AIM, can be viewed as the reasoning tasks and can be adapted to the multi-agent debate, following a similar setting in [3]. Specifically, we can sample different solutions and assign each solution to one debater, and let the moderator select the solution based on the debate. The comprehensive evaluation on more multi-agent scenarios demonstrates that our method is general and promising in reducing communication cost in different scenarios.
>
> [1] AgentDropout: Dynamic Agent Elimination for Token-Efficient and High-Performance LLM-Based Multi-Agent Collaboration
>
> [2] Cut the Crap: An Economical Communication Pipeline for LLM-based Multi-Agent Systems
>
> [3] Debating with more persuasive LLMs leads to more truthful answers, ICML 2024
>
> [4] Improving factuality and reasoning in language models through multiagent debate.
>
> [5] Exploring large language models for communication games: An empirical study on werewolf
>
> [6] A Survey on Large Language Model-Based Game Agents
>
> [7] Natural plan: Benchmarking LLMs on natural language planning
>
> [8] TravelPlanner: A Benchmark for Real-World Planning with Language Agents

---

> > ### Comment · Reviewer_w2uu · 2025-11-28
> >
> > I thank the authors for their detailed response. The additional discussion on decomposing multi-agent communication into atomic one-directional processes helps clarify the intended scope and potential extensibility of HANDRAISER. However, my main concern remains: the current experiments are still limited to task-specific, single-listener settings that resemble user–agent interaction, and there is no empirical validation in genuinely open, multi-directional multi-agent environments (e.g., symmetric interruption among multiple agents). Therefore, while I find the idea interesting and the rebuttal informative, I maintain my overall assessment and score.

---

> ### Author Response · Authors · 2025-12-03
>
> We thank the reviewer for engaging in further discussion. As suggested, we conducted additional experiments to investigate whether our proposed interruptible communication mechanism can also work in other complex scenarios. We extended our Meeting Scheduling Task to the multi-agent symmetric interruptible communication via
>
> (i) **Multi-directional, open interruption**: all 3 agents (two planner agents and one traveler agent) can all interrupt the others, instead of the single-directional interruption.
>
> (ii) **Symmetric interruption**: When the planner agent speaks to the travel agent, the travel agent can interrupt its message; similarly, when the traveler agent responds to the planner agent, the planner agent can also interrupt it.
>
> We directly apply our 70B HandRaise to the decomposed communication process (e.g., Planner 1 → Traveler, Traveler→ Planner 1 and Planner1 → Traveler, Traveler → Planner2) ***without additional finetuning*** . We use the decoding settings and the same baselines as we described in our paper.  All three agents are backboned on the 70B Llama-3 model. The results are shown below:
>
> |  | Generic (no-interrupt) | Random | Prompting | HandRaiser |
> | --- | --- | --- | --- | --- |
> | Success Rate | 0.45$\pm$0.04 | 0.45$\pm$0.06 | 0.46$\pm$0.09 | 0.45$\pm$0.04 |
> | Cost | 1339.5$\pm$25.67 | 1295.99$\pm$74.33 | 1338.59$\pm$11.48 | 1208.67$\pm$41.49 |
>
> From the results, we can find that our HandRaiser can be **adapted to the multi-agent symmetric interruption without additional finetuning**. It can save the communication cost without performance degradation. This shows the promise of our method in genuinely open, multi-directional multi-agent environments.
>
> However, we also find that while the HandRaiser’s interruption reduces the per-round cost by half (103 → 64 tokens), it also improves the number of communication rounds (13.39→20.29 rounds). Further investigation, such as additional finetuning, can be made to improve the interruption efficiency in these complex scenarios, showcasing the extensibility of our proposed framework.

---

### Official Review · Reviewer_j9eZ · 2025-10-23

**Soundness:** 3
**Presentation:** 3
**Contribution:** 3
**Rating:** 6
**Confidence:** 4

**Summary:**

The paper proposes an interruptible communication framework for multi‑agent systems using large language models (LLMs). Current agent communication is verbose and offloads the entire message from the speaker, increasing context and computational costs. Existing methods compress messages from the speaker side but do not allow listeners to control information flow. Inspired by human conversations, the authors introduce a framework in which a listening agent can send an interruption signal to stop the current speaking agent when the listener has enough information or needs clarification.  Initial prompting experiments show that vanilla LLMs tend to interrupt prematurely, so the paper proposes a learning‑based controller called HANDRAISER that predicts optimal interruption points based on estimated future reward (e.g., task performance) and cost (communication tokens). The controller is trained from multi‑agent rollouts with sampled interruptions to estimate the payoff of interrupting versus not interrupting.  HANDRAISER is evaluated on diverse multi‑agent tasks, including 2‑agent text pictionary games, 3‑agent meeting scheduling, and 3‑agent debates. Results indicate that the framework reduces communication cost by around 32 % relative to a baseline while maintaining or improving task performance. The learned interruption behavior generalizes across different agents and tasks.

**Strengths:**

- The paper introduces a novel notion of interruptible communication in multi-agent LLM systems, shifting the control of information flow from the speaker to the listener. This is a fresh angle compared to prior work that focuses on compressing speaker output.
- The HANDRAISER controller is derived from a principled objective that trades off future task reward versus communication cost, and is trained on sampled rollouts to estimate the payoff of interruptions. The authors perform ablations to show that naive prompting-based interrupts make LLMs overconfident and the learning approach improves efficiency.
- Efficient communication is a practical bottleneck in multi-agent systems using large LLMs. Demonstrating a ~32 % reduction in communication cost on several tasks while maintaining or improving performance suggests the method could meaningfully reduce computational cost for multi-agent coordination.
- The paper is generally well written, with a clear problem statement, illustrative figures of the protocol (e.g., pictionary example), and detailed experimental setup. The authors carefully motivate the need for listener-driven interruptions and describe the training procedure, baselines, and metrics.

**Weaknesses:**

- The experiments are limited to small toy scenarios like text Pictionary, simplified meeting scheduling, and controlled debates. It would strengthen the paper to include more realistic multi-agent tasks or to scale up the number of agents to demonstrate robustness.
- The work compares mainly to a baseline of no interruptions. It would be valuable to benchmark against existing communication-efficiency techniques such as message compression, summarization, or reinforcement-learning‑based communication protocols.
- The paper claims that vanilla prompting leads to premature interruptions. However, more in‑depth analysis of why this happens and under what conditions HANDRAISER might still make suboptimal decisions would be helpful. For example, does the learned controller sometimes interrupt too late, affecting task quality?
- The proposed controller is trained using specific tasks and reward structures. It is unclear how sensitive the learned policy is to the choice of reward weights or whether it can transfer to other domains without retraining. Some discussion or experiments on generalization would improve the work.

**Questions:**

- Could the authors clarify how the payoff function is defined and whether it requires task‑specific tuning (e.g., weighting communication cost versus task reward)? How sensitive is the method to these hyperparameters?
- Is the HANDRAISER controller trained jointly across all tasks or separately per task? If trained jointly, how does it avoid overfitting to one task's dynamics?
- In the debate scenario, how is the quality of arguments evaluated? Is there any human evaluation to ensure that interruptions do not degrade content quality?
- Have the authors considered combining their approach with speaker‑side compression or summarization methods? A hybrid approach might yield further improvements.

---

> ### Author Response · Authors · 2025-11-24
>
> We appreciate the reviewer’s constructive comments and suggestions, which can help us strengthen our work. We are glad to see that the reviewer acknowledges the novelty of our interruptible communication and thinks our problem and method are well motivated. We would like to address the reviewer’s concerns as below:
>
> **W1: The experiments are limited to small toy scenarios like text Pictionary, simplified meeting scheduling, and controlled debates.**
>
> A1: To investigate the effectiveness of our method, we evaluate it not only on the reasoning tasks that most multi-agent communication studies focus on [1,2,3,4], but also on two more realistic multi-agent scenarios: multi-agent games [5,6] and daily meeting scheduling [7,8]. These scenarios cover:
>
> - Text Pictionary (Multi-agent Game): 2 agents with different roles: describer and guesser
> - Meeting Scheduling: 3 agents with similar roles but different private information
> - Multi-agent Debate: 3 agents with two types of roles: 2 debaters and 1 moderator
>
> Other math and code reasoning tasks can be adapted to the multi-agent debate, following a similar setting in [3]. The comprehensive evaluation of these multi-agent scenarios demonstrates that our method is general and promising in reducing communication cost in different scenarios.
>
> Meanwhile, we are searching for other types of multi-agent tasks to include. If there are suggestions on the concrete multi-agent tasks, we are happy to investigate and adapt our method.
>
> [1] AgentDropout: Dynamic Agent Elimination for Token-Efficient and High-Performance LLM-Based Multi-Agent Collaboration
>
> [2] Cut the Crap: An Economical Communication Pipeline for LLM-based Multi-Agent Systems
>
> [3] Debating with more persuasive LLMs leads to more truthful answers, ICML 2024
>
> [4] Improving factuality and reasoning in language models through multiagent debate.
>
> [5] Exploring large language models for communication games: An empirical study on werewolf
>
> [6] A Survey on Large Language Model-Based Game Agents
>
> [7] Natural plan: Benchmarking LLMs on natural language planning
>
> [8] TravelPlanner: A Benchmark for Real-World Planning with Language Agents
>
> ---
>
> **W2: It would be valuable to benchmark against existing communication-efficiency techniques such as message compression, summarization, or reinforcement-learning‑based communication protocols. & Q4: combining their approach with speaker‑side compression or summarization methods?**
>
> A2: Thank you for this valuable comment. We have one single speaker-oriented baseline, which is named as **Concise** in Table 1. From the results, we can find that it can also reduce the communication cost, but it will degrade the task performance on tasks such as Meeting Scheduling.
>
> Besides, we also want to highlight that our listener-oriented interruption is orthogonal to the speaker-oriented methods and the communication topology optimization. From the speaker’s side, it can take different methods (prompt-based / RL-based summarization) to compress its message. The listener only cares about the final content it receives from the speaker to decide when to interrupt.
>
> Furthermore, as suggested by the reviewer, we also conduct additional experiments that combine HandRaiser with the speaker-oriented compression (Concise Speaker). The results are shown below. We can find that, on average, the communication cost can be further reduced by a compressed speaker’s message. However, we also find that in cases, such compression will hurt the task performance, similar to the trend we observe in Baseline vs. Concise in Table 1. We think that a more advanced speaker compression method can mitigate such issues.
>
> |  |  | Textual Pictionary |  | Meeting Schedule |  | MMLU-Pro-Debate |  |
> | --- | --- | --- | --- | --- | --- | --- | --- |
> |  |  | SR | Cost | SR | Cost | SR | Cost |
> | Llama-8B | HandRaiser | 0.743 | 262 | 0.307 | 1042 | 0.583 | 782 |
> |  | HandRaiser + Concise Speaker | 0.760 | 253 | 0.273 | 1021 | 0.563 | 763 |
> | Llama-8B | HandRaiser | 0.790 | 294 | 0.447 | 1010 | 0.657 | 806 |
> |  | HandRaiser + Concise Speaker | 0.780 | 279 | 0.467 | 1010 | 0.653 | 794 |

---

> > ### Author Response · Authors · 2025-11-24
> >
> > **W3: More in‑depth analysis of why this happens and under what conditions HANDRAISER might still make suboptimal decisions would be helpful. For example, does the learned controller sometimes interrupt too late, affecting task quality?**
> >
> > A3: Thanks for these valuable suggestions on the failure mode analysis. During our experiments, we found that our HandRaiser is easy to fail in the following scenarios:
> >
> > - *Interrupt too early.* E.g., in Text Pictionary, HandRaiser interrupts right after the describer says “the previous answer is incorrect”, making it unable to get more detailed information for correcting the answer.
> > - *Unsuitable segmentation of chunks*. E.g., in Meeting Scheduling, when saying the available time slots, one chunk may only contain the starting time, making HandRaiser unaware of the ending time and interrupt incorrectly.
> > - *Interrupt but ask for confirmation*. E.g., HandRaisers interrupts but asks the speaker to continue speaking, e.g., “It seems that your statement is incomplete.”, instead of directly responding based on the current partial message.
> >
> > We are conducting a more detailed failure mode analysis and will update with more quantitative results.
> >
> > ---
> >
> > **W4: The proposed controller is trained using specific tasks and reward structures. It is unclear how sensitive the learned policy is to the choice of reward weights or whether it can transfer to other domains without retraining. Some discussion or experiments on generalization would improve the work.**
> >
> > A4: We conducted several generalization studies in Section 4.4 and showed the results in Table 2. We train HandRaiser on one task and directly evaluate it on the other two tasks. Table 2 shows that when transferring to new tasks, the learned interruption behavior can outperform the non-interruptible baseline (Generic) with a lower cost and a comparable performance. We also find that when trained and evaluated on the same task, its performance can be further improved because of a better payoff estimation on this specific task. This is somewhat expected as the optimal timing for interruption is inherently task-specific.
> >
> > ---
> >
> > **Q1: How is the payoff function defined, and whether it requires task‑specific tuning (e.g., weighting communication cost versus task reward)? How sensitive is the method to these hyperparameters?**
> >
> > A1: The payoff consists of two parts: the communication cost $Cost$ and the task performance $Perf$. Instead of using a coefficient to balance the weight between the cost and performance, we define a positive payoff as a negative cost increase and a non-negative performance increase compared with the non-interruptible baseline (as described in Line 185). We label interruption points with a positive payoff as label 1 and leave the other as label 0, and optimize the model towards the label=1 interruption (Lines 226-234). This training objective is more robust than learning an exact payoff value and avoiding the hyperparameter tuning on the coefficient. Sorry for the confusion, and we have revised our **Implementation Details** of Section 4.2 to clarify this.
> >
> > ---
> >
> > **Q2: Is the HANDRAISER controller trained jointly across all tasks or separately per task? If trained jointly, how does it avoid overfitting to one task's dynamics?**
> >
> > A2: Thank you for raising this point. In our main experiment, we train the HandRaiser on each task and evaluate its performance on this task (results in Table 1). We also evaluate the cross-task transferring in Table 2. We didn’t jointly train all three tasks because of the differences in each task’s dynamics. We have added the corresponding details in **Implementation Details of Section 4.2.**
> >
> > ---
> >
> > **Q3: In the debate scenario, how is the quality of arguments evaluated? Is there any human evaluation to ensure that interruptions do not degrade content quality?**
> >
> > A3: In multi-debate scenarios, the task performance is evaluated by the correctness of the answer chosen by the moderator. The moderator should interrupt appropriately to make sure it gets the necessary message from the debater for judgment. In other words, the content quality of the interrupted debater’s message is assessed by the moderator’s performance, which can save human efforts in evaluation.

---

> ### Comment · Reviewer_j9eZ · 2025-11-24
>
> Thanks for the clarification. I will keep my score.

---

> > ### Author Response · Authors · 2025-11-25
> >
> > We appreciate your careful review and are glad that our clarifications addressed your concerns. Any further discussions are more than welcome.

---

### Official Review · Reviewer_kzej · 2025-10-30

**Soundness:** 4
**Presentation:** 2
**Contribution:** 2
**Rating:** 6
**Confidence:** 3

**Summary:**

This paper addresses the problem of communication inefficiency in LLM-based multi-agent systems, where verbose messages increase computational costs and context overload. Instead of the common "speaker-oriented" approach of compressing messages, the authors propose a novel "listener-oriented" framework where listener agents can interrupt speakers. The authors first demonstrate that naive, prompting-based interruption fails because LLMs are "overconfident" and interrupt prematurely, often harming task performance. To solve this, they propose HANDRAISER, a model trained to predict optimal interruption points. This model is finetuned on data labeled by estimating the future task reward and communication cost of potential interruptions via tree-based sampling. Experiments across three distinct multi-agent tasks, including Text Pictionary, Meeting Scheduling, and MMLU-Pro Debate, show that HANDRAISER significantly reduces communication costs by 23-49% while maintaining or improving task success rates compared to both non-interruptible and naive interruption baselines.

**Strengths:**

The paper's primary strength is its novel and intuitive "listener-oriented" interruption framework, which is a departure from standard speaker-side compression. A significant contribution is the paper's clear diagnosis of why naive, prompting-based interruption fails, providing strong empirical evidence that LLMs are "overconfident" and interrupt prematurely, often to their own detriment. This core finding is validated through a thorough evaluation on three diverse multi-agent tasks, demonstrating that the learned HANDRAISER model significantly outperforms strong baselines.

**Weaknesses:**

1. The data-labeling process relies on tree sampling to estimate the future cost and performance of an interruption. However, these future rollouts are conducted using a random interruption policy. This seems like a potentially high-variance and inaccurate estimator for the true value of an interruption, which should depend on optimal or near-optimal future actions, not random ones. But the paper did not provide further details of the choice or analyze what impact it will have.

2. The paper claims there are reductions in "communication cost", usually measured in tokens, and "latency". However, the proposed framework introduces a new computational cost. The listener must run an inference step for the HANDRAISER model at every received chunk. This inference overhead (network + compute altogether) is not measured. In a real-world scenario, it's not very clear whether running 10+ small classifications is actually faster or cheaper than simply processing 30 extra tokens from the speaker.

3. The ablation study in Table 2 indicates that the learned behavior can be used on other tasks, however the performance and cost reductions of different tasks are always lower than those of the original training tasks.  This indicates that the acquired "interruption" skill is highly task-specific and not a universally applicable behavior, thus constraining the method's "plug-and-play" functionality in novel, unfamiliar contexts.

**Questions:**

See above

---

> ### Author Response · Authors · 2025-11-24
>
> We thank the reviewer for their valuable comments, which help us strengthen our work. We are very glad to hear that you find our listener-oriented interruption framework novel and effective in multi-agent communication. We would like to address several concerns you mentioned and welcome your further discussion.
>
> **W1: Further details on the random policy in tree sampling in terms of its potentially high variance and inaccurate estimation for the true value of an interruption.**
>
> A1: Thank you for raising this important point about the tree sampling procedure. We have added one paragraph in our paper to explain the random policy we use in the rollout (**in Appendix C.1 The impact of rollout policies.**). There are three main considerations:
>
> - **Only the symbol (positive / negative) of the payoff is used for training.** As described in Lines 231-233, instead of using the accurate value of the interruption as the training signal, we use the comparison result between the interruption and no interruption. If one interruption has a positive payoff (lower cost and no degraded task performance), it will be labeled as 1. This is because we're not trying to optimize towards the best interruption behavior. Instead, we are gathering training signals that will improve with no interruption. A random policy, while potentially higher variance in the exact payoff value, provides an estimate on whether the interruption can outperform no interruption without a prior hypothesis of the policy.
> - **Preliminary experiments on other policies, such as chain-of-thought interruption, show no significant difference in performance.** We found that this is because, for the same interruption point, the payoff estimation based on a CoT-policy rollout usually has the same training label (0 or 1) as a random rollout.
> - **The random policy is more computationally efficient**. Compared with other policies, the random policy is more efficient for the multi-agent and multi-turn tree sampling.
>
> ---
>
> **W2: This inference overhead (network + compute altogether) is not measured. In a real-world scenario, it's not very clear whether running 10+ small classifications is actually faster or cheaper than simply processing 30 extra tokens from the speaker.**
>
> **A2:** Thank the reviewer for raising the important discussion on real communication cost. We have added a new **Section 3.3 Computation complexity and cost** in our manuscript.
>
> - **Extra computation of encoding introduced by the interruption can be ignored, compared with the decoding cost.** First, the KV cache can be used to avoid repeated computations in encoding previous chunks. Second, the decode phase is significantly more computationally expensive than the encode phase (prefill), especially when the batch size is 1. The main reason is that *the decode phase is memory-bound*: while the computation cost of an input token and an output token is theoretically similar, the decode phase needs to reload model weights from the GPU global memory for each generated token. As an example, popular API providers with advanced batching strategies and parallel architectures, such as OpenAI, still set the price of output tokens 4~10 times higher than the input tokens, indicating a significant computation gap between the input and output.
> - **Generation cost introduced by the number of interruption tokens.** A smaller chunk size leads to more frequent interruption requests, leading to more interruption tokens. Suppose the chunk size is $l_c$ and the length of the full message is $L$. If no interruption is made, then the interruption mechanism will increase $L / l_c$ additional cost for the interruption cost. If an interruption happens (at least one chunk is reduced), then reduced tokens can make up for the interruption token $L / l_c < l_c ⇒ l_c < \sqrt{L}$ . *Therefore, we can select a chunk size larger than $\sqrt{L}$ so that the interruption can benefit the communication cost in the worst case.* In our experiments, the chunk size is 16 (Line 299). The full length of a single message is 83.4 on Text Pictionary, 40.12 on Meeting Scheduling, and 142.79 on Multi-agent Debate, which are all smaller than $16^2=256$.
> - **LLM client and server are usually hosted on the same local network under the benchmark scenarios.** The network latency is negligible (often <1ms). This is also a common practice in most benchmark work, such as vLLM and SGLang.
>
> To conclude, the compute cost and latency are dominated by the compute cost of the newly generated interruption tokens. We can select an appropriate chunk size to ensure that the interruption can benefit the communication in the worst-case scenario.

---

> > ### Author Response · Authors · 2025-11-24
> >
> > **W3. The performance and cost reductions of different tasks are always lower than those of the original training tasks.**
> >
> > A3: We agree that if trained on the specific task, the interruption skill will be better because of a more accurate estimation of the payoff. *This is somewhat expected as the optimal timing for interruption is inherently task-specific.* However, Table 2 also shows that the interruption skill learned from the other tasks can outperform the generic baseline (without interruption). This indicates that, when transferred to novel, unfamiliar contexts without retraining, HandRaiser can also benefit communication at a lower cost with a comparable or better task performance. And this interruption behavior can be further improved if trained on the task-specific payoff.

---

> > > ### Comment · Reviewer_kzej · 2025-11-24
> > >
> > > Thank you for addressing my concerns. I will maintain my positive score.

---

> > > > ### Author Response · Authors · 2025-11-25
> > > >
> > > > We appreciate your careful review and are glad that our clarifications addressed your concerns. Any further discussions are more than welcome.

---

### Official Review · Reviewer_w8hb · 2025-11-01

**Soundness:** 2
**Presentation:** 2
**Contribution:** 2
**Rating:** 4
**Confidence:** 2

**Summary:**

This work addresses the issues of "poor adaptability in speaker-oriented compression" and "LLMs' tendency to be overconfident when interrupting directly" in LLM-based multi-agent communication. It proposes an interruptible communication framework, HANDRaiser, which allows the listener to receive the speaker’s message in fixed-sized chunks, estimates future cost and performance via tree sampling to identify reasonable interruption points, and learns the optimal interruption timing through supervised fine-tuning.

**Strengths:**

1. The paper proposes an interruptible multi-agent communication framework that allows the listener to actively interrupt the speaker during the conversation. This is an important improvement over the traditional “wait-to-finish” model. Introducing the human-like “interruption” mechanism into LLM-based multi-agent systems is both realistic and inspiring.
2. The motivation and problem scope are clear: the paper clearly identifies a practical and timely bottleneck in multi-agent LLM systems—verbosity leading to context overload and computational cost. The focus on listener-initiated interruption is well-justified both conceptually and empirically.

**Weaknesses:**

1. The method is computationally complex. The tree sampling and trajectory annotation processes incur high computational costs, especially in multi-agent, multi-turn settings, which may limit its deployment in real-world systems.
2. The task setup is somewhat idealized: only one agent is allowed to interrupt in the experiments, avoiding the complex scenario of “cascading interruptions.” However, this may not hold in real multi-agent environments. All tasks have clear termination conditions (e.g., guessing the word, scheduling meetings, voting), and it remains unclear whether the method applies to open-domain tasks.

**Questions:**

1. What happens in scenarios where multiple agents are allowed to interrupt? Have you considered, simulated, or conducted rollouts in settings with potential cascading or conflicting interruptions?
2. Would the authors consider conducting human evaluations of communication naturalness, coherence, or user satisfaction to further demonstrate the “human-like” nature of the proposed framework?
3. Can the authors provide a specific analysis or empirical results on the computational cost and scalability of the tree sampling process, especially as the number of agents or chunk granularity increases? How does inference time change in longer or more intensive communication protocols?

---

> ### Author Response · Authors · 2025-11-24
>
> We'd like to thank the reviewer for your careful review, as we are happy to hear that you find our motivation and the proposed listener-oriented interruption mechanism important. We hope we can address the reviewer’s concerns and welcome any further discussion.
>
> **W1 Computation complexity and cost for tree sampling**
>
> A1: We agree that tree sampling is expensive for multi-agent systems. However, such a data collection process is a one-time cost for training, and after the model is trained, it only needs to predict one interruption token at inference time. For real-world applications, inference time efficiency and end-to-end latency are arguably much more important, so we believe our method can be applied in real-world applications.
>
> Moreover, such a sampling process can obtain intermediate reward signals without human supervision; thus, it is widely used to reward modeling in LLMs [1,2].
>
> Additionally, to improve the efficiency of this one-time tree sampling process, we alternate between breadth-first and depth-first expansion algorithms, as well as balancing the maximum branch size, rollout quantity, and number of potential interruption points in the simulation to avoid an exponential increase in computation.
>
> [1] ReST-MCTS∗: LLM Self-Training via Process Reward Guided Tree Search
>
> [2] Improve Mathematical Reasoning in Language Models by Automated Process Supervision
>
> ---
>
> **W2. The complex scenario of “cascading interruptions.” Clear termination conditions may not hold in open-domain tasks. & Q1: Scenarios where multiple agents are allowed to interrupt, such as cascading or conflicting interruptions**
>
> A2: We thank the reviewer for raising the important discussion on other complex scenarios. We would like to address these concerns here and have added this discussion in the **Appendix B Discussion** in our revised manuscript.
>
> **More complicated interruption patterns:** As a first attempt in listener-oriented communication, this paper focuses on the atom communication pattern where only one agent can interrupt the other agents. However, we did consider more complex scenarios such as cascading or conflicting interruptions. Specifically, the communication in multi-agent systems can be categorized into:
>
> - *(i) A fixed speaking order*: the next speaker can interrupt the current speaker
> - *(ii) A free discussion (e.g. group chat)*: all listeners can interrupt the current speaker, then we use first-come-first-serve to decide the interruption (as stated in footnote 2 in Page 3)
>
> We discuss three cases on resolving the complex scenarios. Here, we assume Alice, Bob, and Charlie are three agents.
>
> - *Mutual communication* (Alice → Bob → Alice →… ): decomposed into multiple independent communication Alice → Bob, Bob → Alice, …, each process can use HandRaiser to decide interruption.
> - *Multi-agent communication with a fixed order* (Alice → Bob → Charlie →… ): Similarly, decomposed into independent communication Alice → Bob, Bob → Charlie
> - *Free discussion*: (Alice → Bob while Alice → Charlie, Alice broadcasting to Bob and Charlie): Bob and Charlie independently decide the interruption, while Alice stops at the first interruption point.
>
> These scenarios can be decomposed into the atom communication pattern we investigated in our paper, showing the great promise of HandRaiser. We would like to leave these as our future directions.
>
> **Clear termination conditions.** As with the previous multi-agent LLM system, we focus on goal-oriented tasks [1,2,3,4]. Different from chit-chat[5], there is a clear goal to achieve for the multi-agent system, such as a correct solution for given problems [1,2,3], or a winning condition in games [4]. Therefore, the termination conditions can be set based on: (i) the maximum number of rounds of communication (e.g. 10 in our paper, Line 328) (ii) the status of the goal (e.g., a final answer generated in a pre-defined format, Appendix E Prompt).
>
> **Open-domain tasks**. For open-domain tasks that do not have a single ground-truth answer, we can set up rules to evaluate the status of the goal. For example, in Meeting Scheduling, any schedules that meet the user’s requirements are valid. Therefore, we create a rule-based verifier to evaluate the final schedule (Lines 673-676). Similar approaches can apply to other open-domain tasks.
>
> [1] Improving factuality and reasoning in language models through multiagent debate.
>
> [2] Exploring large language models for communication games: An empirical study on werewolf
>
> [3] Natural plan: Benchmarking LLMs on natural language planning
>
> [4] Language Agents with Reinforcement Learning for Strategic Play in the Werewolf Game
>
> [5] Generative Agents: Interactive Simulacra of Human Behavior

---

> > ### Author Response · Authors · 2025-11-24
> >
> > **Q2: Human evaluations of communication naturalness, coherence, or user satisfaction**
> >
> > A2: We thank the reviewer for this valuable suggestion. The human-like metrics such as naturalness, coherence, and satisfaction are important in human communication, and also for human-agent interaction. However, in agent-agent communication, the effectiveness and efficiency in problem-solving are more important. In our preliminary study, we find that due to the chunk-based segmentation, the interruption often appears in the middle of the sentence, making it less natural and coherent than human communication. We are actively doing more human experiments on these human metrics and will report the new results upon obtaining them.
> >
> > ---
> >
> > **Q3: Analysis or empirical results on the computational cost and scalability of the tree sampling process and inference time in longer or more intensive communication protocols**
> >
> > A3: Thanks for raising this point, and we have added a specific paragraph about the computation cost and scalability of inference in revised Section 3.3.
> >
> > **Inference time in an intensive communication protocol.** HandRaiser’s inference time will be only affected by the number of interruption decisions, which is related to the chunk size. A smaller chunk size leads to more frequent interruption requests, with one token per request.
> >
> > Suppose the chunk size is $l_c$ and the length of the full message is L. If no interruption is made, then the interruption mechanism will increase $L / l_c$ additional cost for the interruption cost. If an interruption happens (at least one chunk is reduced), then reduced tokens can make up for the interruption token $L / l_c < l_c ⇒ l_c < \sqrt{L}$ . Therefore, we can select a chunk size larger than $\sqrt{L}$ to ensure the interruption can always benefit the communication cost.
> >
> > **Computational cost and scalability of Tree sampling:** As the implementation details in Appendix C, we set a maximum branch size of 3 and a rollout number of 10, with a maximum of 10 rounds for each rollout. For one case, we choose at most 10 non-terminating speaking nodes to do rollouts.
> >
> > More generally, suppose there are $n$ agents and each agent takes a turn to speak. For each interruptible agent, we do a separate tree sampling, i.e., we assume only one agent can interrupt in one tree sampling. Therefore, in one round, there are $(n-1)$ messages to be interrupted and create $B$ interrupt branches, resulting in $O(B^n)$ nodes. If we consider T rounds and do $N$ rollouts to estimate all these nodes, it will be $O(TN B^{nT})$, which is very costly. In practice, we can randomly select one message in one round and randomly sample $M$ nodes from the tree to do the rollout, leading to a cost of $O(TNM)$.

---

> > > ### Comment · Reviewer_w8hb · 2025-11-27
> > >
> > > I thank the authors for their response. However, my main concerns remain. First, while inference is efficient, the high computational cost of the tree sampling process during training still poses a significant barrier to scalability and adaptation to new tasks. Second, the proposed strategy for handling cascading interruptions relies on theoretical decomposition without empirical validation, leaving its effectiveness in complex, realistic multi-agent dynamics unproven. Therefore, I maintain my original rating.

---

> ### Author Response · Authors · 2025-12-03
>
> We thank the reviewer for engaging in further discussion. The complexity in multi-agent tree sampling can be avoided by using separate tree sampling for each interruptible agent, making the cost linearly increase with the number of agents. Here, we show how our finetuned HandRaiser based on a one-agent tree sample, can be adapted to multi-agent cascading scenarios.
>
> We extended our Meeting Scheduling Task to the multi-agent cascading interruptible communication via
>
> (i) **2+ interruptible agents**: all three agents (two planner agents and one traveler agent) can all interrupt the others
>
> (ii) **Cascading interruption**: We take the *fixed speaking order* setting as an example, since the *free discussion* setting can be handled with first-come-first-serve. When the planner agent speaks to the travel agent, the travel agent can interrupt its message; similarly, when the traveler agent responds to the planner agent, the planner agent can also interrupt it. E.g., Planner 1 → Traveler → Planner 1 and Planner1 → Traveler → Planner2
>
> We directly apply our 70B HandRaise to the decomposed communication process (e.g., Planner1 → Traveler, Traveler → Planner2) ***without additional finetuning*** . We use the decoding settings and the same baselines as we described in our paper.  All three agents are backboned on the 70B Llama-3 model. The results are shown below:
>
> |  | Generic (no-interrupt) | Random | Prompting | HandRaiser |
> | --- | --- | --- | --- | --- |
> | Success Rate | 0.45$\pm$0.04 | 0.45$\pm$0.06 | 0.46$\pm$0.09 | 0.45$\pm$0.04 |
> | Cost | 1339.5$\pm$25.67 | 1295.99$\pm$74.33 | 1338.59$\pm$11.48 | 1208.67$\pm$41.49 |
>
> From the results, we can find that our HandRaiser can be adapted to the multi-agent symmetric interruption without additional finetuning. It can save the communication cost without performance degradation. This shows the promise of our method in complex, realistic multi-agent dynamics.
>
> However, we also find that while the HandRaiser’s interruption reduces the per-round cost by half (103 → 64 tokens), it also improves the number of communication rounds (13.39→20.29 rounds). Further investigation, such as additional finetuning, can be made to improve the interruption efficiency in these complex scenarios.

---

### Author Response · Authors · 2025-12-03
**Thanks ACs for their hard work and here is a summary comment for the discussion phase**

We sincerely thank the reviewer for the thorough and constructive feedback, which helps us improve the quality of our manuscript.

We are grateful to see that the reviewers think

- Our listener-oriented interruptible communication is novel, intuitive, and insightful.  [Reviewer **w8hb, kzej, j9eZ, w2uu**]
- Human-like “interruption” mechanism into LLM-based multi-agent systems is realistic and inspiring, and is a fresh angle compared to prior speaker-oriented work. [Reviewer **kzej, j9eZ, w8hb]**
- Our motivation is clear and well-justified [Reviewer **w8hb, j9eZ**]
- Our evaluation results across three multi-agent tasks are thorough and significant  [Reviewer **kzej, j9eZ**]

---

During the discussion, we have addressed the concerns mentioned by reviewers, including:

- Concern 1: More complex communication scenarios such as symmetric communication, cascading interruption, and multiple interruptible agents. [Reviewer **w8hb** and **j9eZ**]
    - *Response:*  We show in theory, how our interruption in a single communication process can be adapted to **these complex scenarios by decomposing them into multiple single communication processes** in multi-agent systems
    - *Response:* We also show in practice, our HandRaiser finetuned in a single communication process can **be directly applied to the multi-agent mutual cascading scenario without finetuning**. It can save the communication cost while keeping the task performance
- Concern 2: Computation cost in tree sampling and inference [Reviewer **w8hb** and **kzej**]
    - *Response:* We provide a detailed analysis of the computation cost in tree sampling and show that this one-time cost can be **significantly reduced to a linear relationship** with the number of rollouts and the maximum communication round.
    - *Response:* We also provide a detailed discussion on the additional inference cost introduced by interruption, and show that by **selecting the appropriate chunk size** we can ensure the interruption benefits can overcome the additional cost.
- Concern 3: Combination with speaker-oriented compression [Reviewer **j9eZ**]
    - *Response:* Since our listener-oriented interruption is orthogonal to the speaker-oriented methods and the communication topology optimization, we conduct **additional experiments** to show that the combination can further improve the performance.

---

We really appreciate reviewers’ engagement in discussion, and are happy to see that

- Reviewer **kzej** and **j9eZ** all thought our responses have clarified and addressed their concerns.
- Reviewer **w8hb** and **j9eZ** found our responses have clarified some concerns, but still hold some doubt on the empirical experiments on more complex communication scenarios with multiple interruptible agents (e.g,. symmetric communication or cascading interruption).

***To solve these concerns, we have added additional experiments (as shown below,) but we did not get the chance to get further feedback from them because of the ICLR incident. We hope these additional experiments can be taken into consideration during the AC’s final decision.*** We extended our Meeting Scheduling Task to the multi-agent cascading interruptible communication via

(i) **Multi-agent interruption**: all 3 agents (two planner agents and one traveler agent) can all interrupt the others, instead of the single-directional single-agent interruption.

(ii) **Symmetric communication** with **Cascading interruption**: We take the *fixed speaking order* setting as an example, since the *free discussion* setting can be handled with first-come-first-serve.

We directly apply our 70B HandRaise to the decomposed communication process (e.g., Planner1 → Traveler, Traveler → Planner2) ***without additional finetuning*** . We use the decoding settings and the same baselines as we described in our paper.  All three agents are backboned on the 70B Llama-3 model. The results are shown below:

|  | Generic (no-interrupt) | Random | Prompting | HandRaiser |
| --- | --- | --- | --- | --- |
| Success Rate | 0.45$\pm$0.04 | 0.45$\pm$0.06 | 0.46$\pm$0.09 | 0.45$\pm$0.04 |
| Cost | 1339.5$\pm$25.67 | 1295.99$\pm$74.33 | 1338.59$\pm$11.48 | 1208.67$\pm$41.49 |

We can find that our HandRaiser can be adapted to the multi-agent symmetric interruption without additional finetuning. It can save the communication cost without performance degradation. This shows the promise of our method in genuinely open, multi-directional, multi-agent environments.

---

We have revised our manuscript based on these valuable suggestions (the revised part is highlighted in green), which can be summarized as

- Add computation cost analysis in **Section 3.3**
- Add the discussion of rollout methods in **Appendix C.1**
- Add the discussion of more complex communication scenarios in **Appendix B**
- Clarify the hyperparameter selection and training scope in **Section 4.2**

---

### Meta-Review · Area_Chair_xoZ4 · 2026-01-08

**Summary:**

This research tackles the challenges of "limited adaptability in speaker-oriented compression" and the "overconfidence of LLMs when making direct interruptions" within multi-agent communication systems based on large language models. The authors introduce HANDRaiser, an interruptible communication framework that enables listeners to receive the speaker's message in fixed-size segments. It utilizes tree sampling to estimate future costs and performance, helping to pinpoint appropriate interruption moments. Furthermore, it optimizes the timing of interruptions through supervised fine-tuning.

**Reviewer Concerns:**

The computational cost of the method is high.
The task setting is not quite real.
The data-labeling process is costly.
The ablation studies in the experiment are limited.

**Reviewer Scores:**

The scores of the reviewers are 4,4,6,6, and after rebuttal all the reviewers do not change their scores. This is a boardline paper. It seems that two negative reviewers have read the rebuttal of the authors, but they do not think the responses are sufficient strong to change their scores. Thus, I tend to reject the paper, but my confidence is not high.

---

### Decision · Program_Chairs · 2026-01-26

Reject